# Faster Differentially Private Samplers via Rényi Divergence Analysis of Discretized Langevin MCMC

**Arun Ganesh**
Department of EECS
UC Berkeley[*]
Berkeley, CA 94709
arunganesh@berkeley.edu

**Kunal Talwar**
Apple[†]
Cupertino, CA 95014
ktalwar@apple.com

## Abstract

Various differentially private algorithms instantiate the exponential mechanism, and require sampling from the distribution $\exp(-f)$ for a suitable function $f$. When the domain of the distribution is high-dimensional, this sampling can be computationally challenging. Using heuristic sampling schemes such as Gibbs sampling does not necessarily lead to provable privacy. When $f$ is convex, techniques from log-concave sampling lead to polynomial-time algorithms, albeit with large polynomials. Langevin dynamics-based algorithms offer much faster alternatives under some distance measures such as statistical distance. In this work, we establish rapid convergence for these algorithms under distance measures more suitable for differential privacy. For smooth, strongly-convex $f$, we give the first results proving convergence in Rényi divergence. This gives us fast differentially private algorithms for such $f$. Our techniques and simple and generic and apply also to underdamped Langevin dynamics.

## 1 Introduction

The Exponential Mechanism [McSherry and Talwar, 2007] is a commonly-used mechanism in differential privacy [Dwork and Roth, 2014]. There is a large class of mechanisms in the differential privacy literature that instantiate the Exponential Mechanism with appropriate score functions, use it as a subroutine, or sample from $\exp(-f)$ for some function $f$. This family includes differentially private mechanisms for several important problems, such as PCA [Chaudhuri et al., 2013, Kapralov and Talwar, 2013], functional PCA [Awan et al., 2019], answering counting queries [Hardt and Talwar, 2010], robust regression [Asi and Duchi, 2020], some combinatorial optimization problems [Gupta et al., 2010], $k$-means clustering [Feldman et al., 2009], optimization of dispersed functions [Balcan et al., 2018], convex optimization [Bassily et al., 2014, Minami et al., 2016], Bayesian data analysis [Mir, 2013, Dimitrakakis et al., 2014, Wang et al., 2015, Wasserman and Zhou, 2010, Foulds et al., 2016], linear and quantile regression [Reimherr and Awan, 2019], etc.

Implementing these mechanisms requires sampling from a distribution given by $\exp(-f)$ from some domain $D$, for a suitable score function $f$. When the domain $D$ is finite and small, this sampling is straightforward. Several differentially private mechanisms instantiate the exponential mechanism where $D = \mathbb{R}^d$, in which case this sampling is not straightforward.

Such sampling problems are not new and often occur in statistics and machine learning settings. The common practical approach is to use heuristic MCMC samplers such as Gibbs sampling, which often works well in problems arising in practice. However, given that convergence is not guaranteed, the

---

[*]Part of this work performed while the author was an intern at Google Brain.
[†]Work performed while at Google Brain.

resulting algorithms may not be differentially private. Indeed one can construct simple score functions on the hypercube for which the natural Metropolis chain run for any polynomial time leads to a non-private algorithm [Ganesh and Talwar, 2019]. There are also well-known complexity-theoretic barriers in exactly sampling from $\exp(-f)$ if $f$ is not required to be convex.

Several applications however involve convex functions $f$ and this is the focus of the current work. Indeed this is the problem of sampling from a log-concave distribution, which has attracted a lot of interest. Here, there are two broad lines of work. The classical results in this line of work (e.g. [Applegate and Kannan, 1991, Lovász and Vempala, 2007]) show that given an oracle for computing the function, one can sample from a distribution that is $\varepsilon$-close[3] to the target distribution in time polynomial in $d$ and $\log \frac{1}{\varepsilon}$. Here the closeness is measured in statistical distance. By itself, this does not suffice to give a differentially private algorithm, as DP requires closeness in more stringent notions of distance. The fact that the time complexity is logarithmic in $\frac{1}{\varepsilon}$ however allows for an exponentially small statistical distance in polynomial time. This immediately yields $(\zeta, \delta)$-DP algorithms, and with some additional work can also yield $\zeta$-DP algorithms [Hardt and Talwar, 2010]. Techniques from this line of work can also sometimes apply to non-convex $f$ of interest. Indeed Kapralov and Talwar [2013] designed a polynomial time algorithm for the case of $f$ being a Rayleigh quotient to allow for efficient private PCA.

The runtime of these log-concave sampling algorithms however involves large polynomials. A beautiful line of work has reduced the dependence (of the number of function oracle calls) on the dimension from roughly $d^{10}$ in Applegate and Kannan [1991] to $d^3$ in Lovász and Vempala [2006], Lovász and Vempala [2007]. Nevertheless, the algorithms still fall short of being efficient enough to be implementable in practice for large $d$. A second, more recent, line of work [Dalalyan, 2017, Durmus and Moulines, 2019] have shown that "first order" Markov Chain Monte Carlo (MCMC) algorithms such as Langevin MCMC and Hamiltonian MCMC enjoy fast convergence, and have better dependence on the dimension. These algorithms are typically simpler and more practical but have polynomial dependence on the closeness parameter $\varepsilon$. This polynomial dependence on $\varepsilon$ makes the choice of distance more important. Indeed these algorithms have been analyzed for various measures of distance between distributions such as statistical distance, KL-divergence and Wasserstein distance.

These notions of distance however do not lead to efficient differentially private algorithms (see Appendix F). This motivates the question of establishing rapid mixing in Rényi divergence for these algorithms. This is the question we address in this work, and show that when $f$ is smooth and strongly convex, discretized Langevin dynamics converge in iteration complexity near-linear in the dimension. This gives more efficient differentially private algorithms for sampling for such $f$.

Vempala and Wibisono [2019] recently studied this question, partly for similar reasons. They considered the Unadjusted (i.e., overdamped) Langevin Algorithm and showed that when the (discretized) Markov chain satisfies suitable mixing properties (e.g. Log Sobolev inequality), then the discrete process converges in Rényi divergence to *a* stationary distribution. However this stationary distribution of the discretized chain is different from the target distribution. The Rényi divergence between the stationary distribution and $\exp(-f)$ is not very well-understood [Roberts and Tweedie, 1996, Wibisono, 2018], and it is conceivable that the stationary distribution of the discrete process is *not* close in Rényi divergence to the target distribution and thus may not be differentially private. Thus the question of designing fast algorithms that sample from a distribution close to the distribution $\exp(-f)$ in Rényi divergence was left open.

In this work we use a novel approach to address these questions of fast sampling from $\exp(-f)$ using the discretized Langevin Algorithm. Interestingly, we borrow tools commonly used in differential privacy, though applied in a way that is not very intuitive from a privacy point of view. We upper bound the Rényi divergence between the output of the discrete Langevin Algorithm run for $T$ steps, and the output of the continuous process run for time $T\eta$. The continuous process is known [Vempala and Wibisono, 2019] to converge very quickly in Rényi divergence to the target distribution. This allows us to assert closeness (in Rényi divergence) of the output of the discrete algorithm to the target distribution. This bypasses the question of the bias of the stationary distribution of the discrete process. Moreover, this gives us a differentially private algorithm with iteration complexity near-linear in the

| $f$ is $L$-smooth and | Process | $\eta$ | Iterations |
|---|---|---|---|
| 1-strongly convex | Overdamped | $\tilde{O}\left(\frac{1}{\tau L^4 \ln^2 \alpha} \cdot \frac{\varepsilon^2}{d}\right)$ (Thm 9) | $\tilde{O}\left(\frac{d\tau^2 L^4 \ln^2 \alpha}{\varepsilon^2}\right)$ |
| $B$-Lipschitz | Overdamped | $\tilde{O}\left(\frac{1}{\tau L^4 \ln^2 \alpha} \cdot \frac{\varepsilon^2}{B^2+d}\right)$ (Thm 12) | $\tilde{O}\left(\frac{(B^2+d)\tau^2 L^4 \ln^2 \alpha}{\varepsilon^2}\right)$ |
| 1-strongly convex | Underdamped | $\tilde{O}\left(\frac{1}{\tau L \ln \alpha} \cdot \frac{\varepsilon}{\sqrt{d}}\right)$ (Thm 16) | $\tilde{O}\left(\frac{\sqrt{d}\tau^2 L \ln \alpha}{\varepsilon}\right)$ |

Figure 1: Summary of results. For each family of functions and process (either overdamped or underdamped Langevin dynamics), an upper bound is listed on the step size $\eta$ (and thus a bound on the iteration complexity) needed to ensure the $\alpha$-Rényi divergence between the discrete and continuous processes is at most $\varepsilon$ after time $\tau$. Setting $\alpha = O(\ln(1/\delta)/\zeta), \varepsilon = \zeta/2$ gives that the $\delta$-approximate max divergence is at most $\zeta$, i.e. $(\zeta, \delta)$-differential privacy.

dimension. Our result applies to log-smooth and strongly log-concave distributions. While results of this form may also be provable using methods from optimal transport, we believe that our techniques are simpler and more approachable to the differential privacy community, and may be more easily adaptable to other functions $f$ of interest.

Our approach is general and simple. We show that it can be extended to the *underdamped* Langevin dynamics which have a better dependence on dimension, modulo proving fast mixing for the continuous process. As a specific application, we show how our results lead to faster algorithms for implementing the mechanisms in Minami et al. [2016].

As is common in this line of work, we ignore numerical issues and assume real arithmetic. The results can be translated to the finite-precision arithmetic case by standard techniques, as long as the precision is at least logarithmic in $d$ and $T$. The real arithmetic assumption thus simplifies the presentation without affecting the generality of the results.

## 1.1 Other Related Work

Wang et al. [2015] discuss the issue of privacy when using approximate samplers at length and consider two algorithms. The first one (OPS) that samples approximately from $\exp(-f)$ considers closeness in statistical distance and thus can only be efficient when coupled with the first kind of samplers above, i.e. those that have a logarithmic dependence on the closeness parameter. The second algorithm they analyze is a variant of Stochastic Gradient Langevin Dynamics (SGLD). The algorithm adds additional noise for privacy, and while it is shown to be private for suitable parameters, it does not ensure convergence to the target distribution. Differentially private approximations to SGLD have also been studied in Li et al. [2019]. Note that in contrast, we do not need to modify the Langevin dynamics which ensures convergence as well as privacy.

There is a large body of work on Langevin algorithms and their variants. We refer the reader to the surveys by Roberts and Rosenthal [2004] and Vempala [2005]. There has been a recent spate of activity on analyzing these algorithms and their stochastic variants, under different kinds of assumptions on $f$ and we do not attempt to summarize it here.

## 1.2 Results and Techniques

Our results are summarized in Figure 1. Combined with results from Vempala and Wibisono [2019] on the convergence of the continuous process, the first result gives the following algorithmic guarantee, our main result:

**Theorem 1.** *Fix any $\alpha \geq 1$. Let $R$ be a distribution satisfying $R(x) \propto e^{-f(x)}$ for 1-strongly convex and $L$-smooth $f$ with global minimum at 0. Let $P$ be the distribution arrived at by running discretized overdamped Langevin dynamics using $f$ with step size $\eta = \tilde{O}(\frac{1}{\tau L^4 \ln^2 \alpha} \cdot \frac{\varepsilon^2}{d})$ for continuous time $\tau = O(\alpha \ln \frac{d \ln L}{\varepsilon})$ (i.e. for $\tilde{O}(\frac{\alpha^2 L^4 d}{\varepsilon^2})$ steps) from initial distribution $N(0, I_d)$. Then we have $D_\alpha(P||R), D_\alpha(R||P) \leq \varepsilon$.*

This is the first algorithmic result for sampling from log-smooth and strongly log-concave distributions with low error in Rényi divergence without additional assumptions. In particular, if for $\alpha = 1 + 2\log(1/\delta)/\zeta$ we have $D_\alpha(P||R), D_\alpha(R||P) \leq \zeta/2$, then by Fact 4 we have that $P, R$ satisfy

the divergence bounds of $(\zeta, \delta)$-differential privacy. In turn, given any mechanism that outputs $R, R'$ on adjacent databases satisfying $(\zeta, \delta)$-differential privacy and the strong convexity and smoothness conditions, Theorem 1 and standard composition theorems gives a mechanism that outputs $P, P'$ for these databases such that the mechanism satisfies $(3\zeta, 3\delta)$-differential privacy, $P, P'$ are efficiently sampleable, and $P, P'$ obtain utility guarantees comparable to those of $R, R'$.

All results in Figure 1 are achieved using a similar analysis, which we describe here. Instead of directly bounding the divergence between the discrete and continuous processes, we instead bound the divergence between the discrete processes using step sizes $\eta, \eta/k$. Our resulting bound does not depend on $k$, so we can take the limit as $k$ goes to infinity and the latter approaches the continuous process. Suppose within each step of size $\eta$, neither process moves more than $r$ away from the position at the start of this step. Then by smoothness, in each interval of length $\eta/k$ the distance between the gradient steps between the two processes is upper bounded by $Lr\frac{\eta}{k}$. Our divergence bound thus worsens by at most $D_\alpha(N(0, \frac{2\eta}{k})||N(x, \frac{2\eta}{k}))$ where $x$ is a vector with $\|x\|_2 \leq Lr\frac{\eta}{k}$. The divergence between shifted Gaussians is well-known, giving us a divergence bound.

Of course, since the movement due to Brownian motion can be arbitrarily large, there is no unconditional bound on $r$. Instead, we derive tail bounds for $r$, giving a divergence bound (depending on $\delta$) between the two processes conditioned on a probability $1 - \delta$ event for every $\delta$. We then show a simple lemma which says that conditional upper bounds on the larger moments of a random variable give an unconditional upper bound on the expectation of that random variable. By the definition of Rényi divergence, $\exp((\alpha' - 1)D_{\alpha'}(P||Q))$ is a moment of $\exp((\alpha - 1)D_\alpha(P||Q))$ for $\alpha' > \alpha$, so we can apply this lemma to our conditional bound on $\alpha'$-Rényi divergence to get an unconditional bound on $\alpha$-Rényi divergence via Jensen's inequality.

Finally, since our analysis only needs smoothness, the radius tail bound, and the fact that the process is a composition of gradient steps with Gaussian noise, our analysis easily extends to sampling from Lipschitz rather than strongly convex functions and analyzing the underdamped Langevin dynamics.

As an immediate application, we recall the work of Minami et al. [2016], who give a $(\zeta, \delta)$-differentially private mechanism that (approximately) samples from a Gibbs posterior with a strongly log-concave prior, for applications such as mean estimation and logistic regression. Their iteration complexity of $\tilde{O}(d^3/\delta^2)$ proved in Minami et al. [2016, Prop. 13] gets improved to $\tilde{O}(d/\zeta^4)$ using our main result. We note that the privacy parameters in $(\zeta, \delta)$-DP that one typically aims for are $\zeta$ being constant, and $\delta$ being negligible. However, it is still an interesting open problem to improve the iteration complexity's dependence on $\zeta$.

We start with some preliminaries in Section 2. We prove a "one-sided" version of Theorem 1 in Section 3 and Section 4, extend this analysis to prove Theorem 1 in Section 5, and prove the result for the underdamped case in Section 6. We discuss future research directions in Section 7. Most proofs are deferred to the supplementary materials.

## 2 Preliminaries

### 2.1 Langevin Dynamics and Basic Assumptions

For the majority of the paper we focus on the overdamped Langevin dynamics in $\mathbb{R}^d$, given by the following stochastic differential equation (SDE):

$$\mathrm{d}x_t = -\nabla f(x_t)\mathrm{d}t + \sqrt{2}\mathrm{d}B_t,$$

Where $B_t$ is a standard $d$-dimensional Brownian motion. Under mild assumptions (such as strong convexity of $f$), it is known that the stationary distribution of the SDE is the distribution $p$ satisfying $p(x) \propto e^{-f(x)}$. Algorithmically, it is easier to use the following discretization with *steps* of size $\eta$:

$$\mathrm{d}x_t = -\nabla f(x_{\lfloor \frac{t}{\eta} \rfloor \eta})\mathrm{d}t + \sqrt{2}\mathrm{d}B_t,$$

i.e., we only update the gradient used in the SDE at the beginning of each step. Restricted to the position at times that are multiples of $\eta$, equivalently:

$$x_{(i+1)\eta} = x_{i\eta} - \eta\nabla f(x_{i\eta}) + \xi_i.$$

Where $\xi_i \sim N(0, 2\eta I)$ are independent samples. Throughout the paper, when we refer to the result of running a Langevin dynamics for *continuous time* $t$, we mean the distribution $x_t$, *not* the distribution

$x_{t\eta}$. When the iteration complexity (i.e. number of steps) is of interest, we may refer to running a Langevin dynamics for continuous time $T\eta$ equivalently as the result of running it for $T$ steps (of size $\eta$).

A similarly defined second order process is the underdamped Langevin dynamics, given by the following SDE (parameterized by $\gamma, \mu > 0$):

$$\mathrm{d}v_t = -\gamma v_t \mathrm{d}t - \mu \nabla f(x_t)\mathrm{d}t + \sqrt{2\gamma\mu}\mathrm{d}B_t, \qquad \mathrm{d}x_t = v_t \mathrm{d}t.$$

Again, under mild assumptions it is known that the stationary distribution of this SDE is the distribution $p$ satisfying $p(x) \propto e^{-(f(x) + \|v\|_2^2/2\mu)}$, so that the marginal on $x$ is as desired. Algorithmically, it is easier to use the following discretization:

$$\mathrm{d}v_t = -\gamma v_t \mathrm{d}_t - \mu \nabla f(x_{\lfloor \frac{t}{\eta} \rfloor \eta})\mathrm{d}t + \sqrt{2\gamma\mu}\mathrm{d}B_t, \qquad \mathrm{d}x_t = v_t \mathrm{d}t. \tag{1}$$

In the majority of the paper we consider sampling from distributions given by $m$-strongly convex, $L$-smooth functions $f$. To simplify the presentation, we also assume $f$ is twice-differentiable, so these conditions on $f$ can be expressed as: for all $x$, $mI \preccurlyeq \nabla^2 f(x) \preccurlyeq LI$. We make two additional simplifying assumptions: The first is that the stationary point of $f$ is at 0, as if $f$'s true stationary point is $x^* \neq 0$, we can sample from $g(x) := f(x - x^*)$ and then shift our sample by $x^*$ to get a sample from $f$ instead ($x^*$ can be found using e.g. gradient descent). The second is that $m = 1$, as if $m \neq 1$, we can sample from $g(x) = f(\frac{1}{\sqrt{m}}x)$ and rescale our sample by $\sqrt{m}$ instead.

## 2.2 Rényi Divergence

**Definition 2** (Rényi Divergence). *For $0 < \alpha < \infty$, $\alpha \neq 1$ and distributions $\mu, \nu$, such that $supp(\mu) = supp(\nu)$ the $\alpha$-Rényi divergence between $\mu$ and $\nu$ is*

$$D_\alpha(\mu||\nu) = \frac{1}{\alpha - 1}\ln \int_{supp(\nu)} \frac{\mu(x)^\alpha}{\nu(x)^{\alpha-1}}dx = \frac{1}{\alpha - 1}\ln \mathbb{E}_{x\sim\nu}\left[\frac{\mu(x)^\alpha}{\nu(x)^\alpha}\right].$$

*The $\alpha$-Rényi divergence for $\alpha = 1$ (resp. $\infty$) is defined by taking the limit of $D_\alpha(\mu||\nu)$ as $\alpha$ approaches 1 (resp. $\infty$) and equals the KL divergence (resp. max divergence).*

Rényi divergence is a standard notion of divergence in information theory, and Rényi divergence bounds translate to differential privacy guarantees:

**Definition 3** (Approximate Differential Privacy). *The $\delta$-approximate max divergence between distributions $\mu, \nu$ is defined as:*

$$D_\infty^\delta(\mu||\nu) = \max_{S \subseteq supp(\mu): \Pr_{x\sim\mu}[X\in S]\geq\delta} \left[\ln \frac{\Pr_{x\sim\mu}[x \in S] - \delta}{\Pr_{x\sim\nu}[x \in S]}\right]$$

**Fact 4** ([Mironov, 2017, Proposition 3]). *For $\alpha > 1$ if $\mu, \nu$ satisfy $D_\alpha(\mu||\nu) \leq \zeta$, then for $0 < \delta < 1$:*

$$D_\infty^\delta(\mu||\nu) \leq \zeta + \frac{\ln(1/\delta)}{\alpha - 1}.$$

## 3 Langevin Dynamics with Bounded Movements

As a first step, we analyze the divergence between the discrete and continuous processes conditioned on the event $\mathcal{E}_r$ that throughout each step of size $\eta$ they stay within a ball of radius $r$ around their location at the start of the step. We will actually analyze the divergence between two discrete processes with steps of size $\eta$ and $\eta/k$ respectively, and obtain a bound on their divergence independent of $k$. The former is exactly the discrete Langevin dynamics with step size $\eta$. The limit of the latter, as $k$ goes to infinity, is the continuous Langevin dynamics. Thus the same bound applies to the divergence between the discrete and continuous processes. We set up discretized overdamped Langevin dynamics with step sizes $\eta, \eta/k$ as random processes which record the position at each time that is a multiple of $\eta/k$.

Let $x_t$ denote the position of the chain using step size $\eta$ at continuous time $t$, and $x_t'$ denote the position of the chain using step size $\eta/k$ at time $t$. If $\mathcal{E}_r$ does not hold at time $t^*$ (more formally, if

$\max_{t\in[0,t^*]}\|x_t - x_{\lfloor t/\eta\rfloor\eta}\|_2 > r)$, we will instead let $x_t = \perp$ for all $t \geq t^*$. We want to bound the divergence after $T$ steps of length $\eta$, i.e. the divergence between the distributions of $x_{T\eta}$ and $x'_{T\eta}$. Let $X_{0:j}$ denote the distribution of $\{x_{i\eta/k}\}_{0\leq i\leq j}$, and define $X'_{0:j}$ analogously. By the post-processing property of Rényi DP (i.e. data processing inequality; see Fact 18 in supplementary materials), it suffices to bound the divergence between $X_{0:Tk}$ and $X'_{0:Tk}$. Note that we can sample from $X_{0:Tk}$ (resp $X'_{0:Tk}$) by starting with $\{x_0\}$ (resp $\{x'_0\}$) sampled from the distribution $X_0$ from which we start the Langevin dynamics, and applying the following randomized update $Tk$ times:

- To draw a sample from $X_{0:Tk}$, given a sample $\{x_{i\eta/k}\}_{0\leq i\leq j}$ from $X_{0:j}$:
  - If $x_{j\eta/k} = \perp$ append $x_{(j+1)\eta/k} = \perp$ to $\{x_{i\eta/k}\}_{0\leq i\leq j}$ to get a sample from $X_{0:j+1}$.
  - Otherwise, append $x_{(j+1)\eta/k} = x_{j\eta/k} - \frac{\eta}{k}\nabla f(x_{\lfloor j/k\rfloor\eta}) + \xi_j$, where $\xi_j \sim N(0, \frac{2\eta}{k}I_d)$ to get a sample from $X_{0:j+1}$. Then if $\|x_{(j+1)\eta/k} - x_{\lfloor(j+1)/k\rfloor\eta}\|_2 > r$ (i.e. $\mathcal{E}_r$ no longer holds) replace $x_{(j+1)\eta/k}$ with $\perp$.

  We will denote this update by $\psi$. More formally, $\psi$ is the map from distributions to distributions such that $X_{0:j+1} = \psi(X_{0:j})$.

- To draw a sample from $X'_{0:Tk}$, we instead use the update $\psi'$ that is identical to $\psi$ except $\psi'$ uses the gradient at $x'_{j\eta/k}$ instead of $x'_{\lfloor j/k\rfloor\eta}$.

We now have $X_{0:Tk} = \psi^{\circ Tk}(X_0)$ and $X'_{0:Tk} = (\psi')^{\circ Tk}(X_0)$. The divergence between $\psi(X), \psi'(X)$ for some $X$ can be bounded by the Rényi divergence between two Gaussians, and then a composition theorem gives the following bound on the divergence between the final tuples.

**Lemma 5.** *For any $L$-smooth $f$, any initial distribution $X_0$ over $x_0, x'_0$, and the distributions over tuples $X_{0:Tk}, X'_{0:Tk}$ as defined above, we have:*

$$D_\alpha(X_{0:Tk}\|X'_{0:Tk}), D_\alpha(X'_{0:Tk}\|X_{0:Tk}) \leq \frac{T\alpha L^2 r^2\eta}{4}.$$

Note that if we are running the process for continuous time $\tau$, then $T = \tau/\eta$. $r$ will end up being roughly proportional to $\sqrt{\eta}$, so the above bound is then roughly proportional to $\eta$.

## 4 Removing the Bounded Movement Restriction

In this section, we will prove the following "one-sided" version of Theorem 6:

**Theorem 6.** *Fix any $\alpha \geq 1$. Let $R$ be a distribution satisfying $R(x) \propto e^{-f(x)}$ for 1-strongly convex and $L$-smooth $f$ with global minimum at 0. Let $P$ be the distribution arrived at by running discretized overdamped Langevin dynamics using $f$ with step size $\eta = \tilde{O}(\frac{1}{\tau L^4 \ln^2\alpha} \cdot \frac{\varepsilon^2}{d})$ for continuous time $\tau = \alpha\ln\frac{d\ln L}{\varepsilon}$ (i.e. for $\tilde{O}(\frac{\alpha^2 L^4 d}{\varepsilon^2})$ steps) from initial distribution $N(0, \frac{1}{L}I_d)$. Then we have $D_\alpha(P\|R) \leq \varepsilon$.*

To remove the assumption that the process never moves more than $r$ away from its original position within each step of size $\eta$, we give a tail bound on the maximum value $r$ that the process moves within one of these steps.

**Lemma 7.** *Let $c$ be a sufficiently large constant. Let $\eta \leq \frac{2}{L+1}$ and let $X_0$ be an initial distribution over $\mathbb{R}^d$ satisfying that for all $\delta > 0$,*

$$\Pr_{x\sim X_0}\left[\|x\|_2 \leq \frac{c}{2\sqrt{\eta}}\left(\sqrt{d} + \sqrt{\ln(T/\delta)}\right)\right] \geq 1 - \frac{\delta}{4(T+1)}. \tag{2}$$

*Let $x_t$ be the random variable given by running the discretized overdamped Langevin dynamics starting from $X_0$ for continuous time $t$. Similarly, let $x'_t$ be the random variable given by running continuous overdamped Langevin dynamics starting from $X_0$ for continuous time $t$. Then with probability at least $1 - \delta$ over the path $\{x_t : t \in [0, T\eta]\}$ and with probability at least $1 - \delta$ over the path $\{x'_t : t \in [0, T\eta]\}$:*

$$\forall t \leq T\eta : \left\|x_t - x_{\lfloor t/\eta\rfloor\eta}\right\|_2, \left\|x'_t - x'_{\lfloor t/\eta\rfloor\eta}\right\|_2 \leq cL\left(\sqrt{d} + \sqrt{\ln(T/\delta)}\right)\sqrt{\eta}.$$

Intuitively, the $\sqrt{\eta}$ accounts for movement due to Brownian motion, which dominates the movement due to the gradient, and $cL(\sqrt{d} + \sqrt{\ln(T/\delta)})$ is a tail bound on norm of the gradient by smoothness. This gives us a bound on the Rényi divergence between the continuous and discrete processes conditioned on a probability $1 - \delta$ event for all $0 < \delta < 1$. By absorbing the failure probability of this event into the probability of large privacy loss in the definition of $(\zeta, \delta)$-differential privacy we can prove iteration complexity bounds matching those in Figure 1 for running discretized overdamped Langevin dynamics with $(\zeta, \delta)$-differential privacy. Since these bounds do not improve on those in the ones derived from our final (unconditional) divergence bounds, we omit the proof here. To prove a Rényi divergence bound, we need to remove the conditioning. We start with the following lemma, which takes bounds on conditional moments and gives an unconditional bound on expectation:

**Lemma 8.** *Let $Y$ be a random variable distributed over $\mathbb{R}_{\geq 0}$ that has the following property (parameterized by positive parameters $\beta, \gamma < 1, \theta > 1 + \gamma$): For every $0 < \delta < 1/2$, there is a probability $\geq 1 - \delta$ event $\mathcal{E}_\delta$ such that $\mathbb{E}\left[Y^\theta | \mathcal{E}_\delta\right] \leq \frac{\beta}{\delta^\gamma}$. Then we have:*

$$\mathbb{E}[Y] \leq \beta^{\frac{1}{\theta}} \left(\gamma^{\frac{1}{1+\gamma}} + \gamma^{-\frac{\gamma}{1+\gamma}}\right)^{\frac{1+\gamma}{\theta}} \left(\frac{\theta(1+\gamma)}{\theta(1+\gamma)-1}\right) \leq \beta^{1/\theta} 2^{2/\theta} \frac{\theta}{\theta-1}.$$

Putting it all together, we get the following lemma:

**Theorem 9.** *For any $1$-strongly convex, $L$-smooth $f$, let $P$ be the distribution of states for discretized overdamped Langevin dynamics with step size $\eta$ and $Q$ be the distribution of states for continuous overdamped Langevin dynamics, both run from any initial distribution $X_0$ satisfying (2) for continuous time $\tau$ that is a multiple of $\eta$ (i.e. for $\tau/\eta$ steps). Then for $\alpha > 1$, $\varepsilon > 0$, if $\eta = \tilde{O}(\frac{1}{\tau L^4 \ln^2 \alpha} \cdot \frac{\varepsilon^2}{d})$ we have $D_\alpha(P||Q), D_\alpha(Q||P) \leq \varepsilon$.*

We provide some high level intuition for the proof here. Plugging Lemma 7 into Lemma 5 gives a bound on roughly the $\alpha'$-Rényi divergence between $P$ conditioned on some probability $1 - \delta_1$ event and $Q$ conditioned on some probability $1 - \delta_2$ event for every $\delta_1, \delta_2$. We apply Lemma 8 once for $P$ and once for $Q$ to remove the conditioning, giving a bound of $\approx \frac{\ln \alpha'}{\alpha'-1}$ on the actual $\alpha'$-Rényi divergence between $P, Q$ if $\eta$ is sufficiently small (as a function of $\alpha'$). Using Jensen's inequality, we can turn this into a bound of $\varepsilon$ on the $\alpha$-Rényi divergence between $P, Q$ for any $\alpha$ if $\alpha'$ is large enough (which in turn requires $\eta$ to be small enough).

We now apply results from Vempala and Wibisono [2019] and the weak triangle inequality for Rényi divergence to get a bound on the number of iterations of discrete overdamped Langevin dynamics needed to achieve $\alpha$-Rényi divergence $\varepsilon$:

**Lemma 10.** *If $R(x) = e^{-f(x)}$ is a probability distribution over $\mathbb{R}^d$ with stationary point $0$ and $f$ is $1$-strongly convex and $L$-smooth, then for all $\alpha \geq 1$ we have:*

$$D_\alpha\left(N\left(0, \frac{1}{L}I_d\right)||R\right) \leq \frac{d}{2}\ln L.$$

It is well-known that $1$-strong convexity of $f$ implies that $p \propto e^{-f}$ satisfies log-Sobolev inequality with constant $1$ (see e.g. Bakry and Émery [1985]). We then get:

**Lemma 11** (Theorem 2, Vempala and Wibisono [2019]). *Fix any $f$ that is $1$-strongly convex. Let $Q_t$ be the distribution arrived at by running overdamped Langevin dynamics using $f$ for continuous time $t$ from initial distribution $Q_0$. Then for $R = e^{-f}$ and any $\alpha \geq 1$:*

$$D_\alpha(Q_t||R) \leq e^{-2t/\alpha} D_\alpha(Q_0||R).$$

Theorem 6 follows from Lemmas 10 and 11 and a weak triangle inequality for Rényi divergences.

## 4.1 Langevin Dynamics with Bounded Gradients

With only a minor modification to the analysis of the strongly convex and smooth case, we can also give a discretization error bound when $f$ is $B$-Lipschitz instead of strongly convex (while still $L$-smooth). We derive a simple tail bound similar to Lemma 7 in the appendix, and then repeat the analysis of Theorem 9 using the new tail bound. This gives:

**Theorem 12.** *For any $B$-Lipschitz, $L$-smooth function $f$, let $P$ be the distribution of states for discretized overdamped Langevin dynamics with step size $\eta$ and $Q$ be the distribution of states for continuous overdamped Langevin dynamics, both run from arbitrary initial distribution for continuous time $\tau$ that is a multiple of $\eta$. Then for $\alpha > 1$, $\varepsilon > 0$, if $\eta = \tilde{O}(\frac{1}{\tau L^4 \ln^2 \alpha} \cdot \frac{\varepsilon^2}{B^2 + d})$ we have $D_\alpha(P\|Q) \leq \varepsilon$.*

## 5 Making The Bound Bi-Directional

In this section, we show that with slight modifications to the proof of Theorem 6, $D_\alpha(P\|R)$ and $D_\alpha(R\|P)$ can be simultaneously bounded, proving Theorem 1. We first need a lemma analogous to Lemma 11 to show that $D_\alpha(R\|Q)$ decays exponentially:

**Lemma 13.** *Fix any $f$ that is $1$-strongly convex. Let $Q_t$ be the distribution arrived at by running overdamped Langevin dynamics using $f$ for continuous time $t$ from initial distribution $Q_0$ such that $-\log Q_0$ is $1$-strongly convex. Then for the distribution $R$ satisfying $R(x) \propto e^{-f(x)}$, any $\alpha > 1$, and any $t$:*

$$D_\alpha(R\|Q_t) \leq e^{-t/\alpha} D_\alpha(R\|Q_0).$$

This proof follows similarly to Lemma 2 in Vempala and Wibisono [2019]. If $D_\alpha(R\|Q_0)$ and $D_\alpha(Q_0\|R)$ were both initially not too large, Lemma 13 along with Lemma 11, Theorem 9 would be enough to arrive at Theorem 1. Unfortunately in general we can't hope for any $Q_0$ to satisfy this, but the following lemmas let us show that for $Q_0 = N(0, I_d)$, $D_\alpha(R\|Q_t)$ and $D_\alpha(Q_t\|R)$ are both not too large for a reasonably small choice of $t$.

**Lemma 14** (Lemma 14, Vempala and Wibisono [2019]). *Fix any $f$ that is $1$-strongly convex. Let $Q_t$ be the distribution arrived at by running overdamped Langevin dynamics using $f$ for continuous time $t$ from initial distribution $Q_0$. Fix any $\alpha_0 > 1$, and let $\alpha_t = 1 + e^{2t}(\alpha_0 - 1)$. Then for the distribution $R$ satisfying $R(x) \propto e^{-f(x)}$:*

$$D_{\alpha_t}(Q_t\|R) \leq \frac{1 - 1/\alpha_0}{1 - 1/\alpha_t} D_{\alpha_0}(Q_0\|R).$$

**Lemma 15.** *Let $Q_0 = N(0, I_d)$. If $R(x) = e^{-f(x)}$ is a probability distribution over $\mathbb{R}^d$ with stationary point $0$ and $f$ is $1$-strongly convex and $L$-smooth, then for all $\alpha \geq 1$ we have:*

$$D_\alpha(R\|Q_0) \leq d \log L.$$

*In addition:*

$$D_{1+1/L}(Q_0\|R) \leq \frac{dL \log L}{2}.$$

Theorem 1 now follows from these lemmas, Theorem 9, Lemma 11 and the weak triangle inequality for Rényi divergence.

## 6 Underdamped Langevin Dynamics

Our approach can also be used to show a bound on the discretization error of *underdamped* Langevin dynamics. We again start by bounding the divergence between two discrete processes with step sizes $\eta$ and $\eta/k$, whose limits as $k$ goes to infinity are the discretized and continuous underdamped Langevin dynamics. Again let $x_t$ denote the position of the chain using step size $\eta$ at continuous time $t$, and $x'_t$ denote the position of the chain using step size $\eta/k$. Let $v_t, v'_t$ denote the same but for velocity instead of position. If e.g. for the first chain we ever have $\|x_{t^*} - x_{\lfloor t^*/\eta \rfloor \eta}\|_2 > r$ we will let $(x_t, v_t)$ equal $\perp$ for all $t \geq t^*$. We want to bound the divergence between the distributions $X_{0:Tk}$ over $\{(x_{i\eta/k}, v_{i\eta/k})\}_{0 \leq i \leq Tk}$ and $X'_{0:Tk}$ over $\{(x'_{i\eta/k}, v'_{i\eta/k})\}_{0 \leq i \leq Tk}$. A sample from $X_{0:Tk}$ or $X'_{0:Tk}$ can be constructed by applying the following operations $Tk$ times to $\{(x_0, v_0)\}$ sampled from an initial distribution $X_0$:

- To construct a sample from $X_{0:Tk}$, given a sample $\{(x_{i\eta/k}, v_{i\eta/k})\}_{0 \leq i \leq j}$ from $X_{0:j}$:
  - If $(x_{j\eta/k}, v_{j\eta/k}) = \perp$ append $(x_{i\eta/k}, v_{i\eta/k}) = \perp$ to $\{(x_{i\eta/k}, v_{i\eta/k})\}_{0 \leq i \leq j}$.

– Otherwise, append $(x_{(j+1)\eta/k}, v_{(j+1)\eta/k})$ where:

$$v_{(j+1)\eta/k} = (1-\gamma\frac{\eta}{k})v_{j\eta/k} - \mu\frac{\eta}{k}\nabla f(x_{\lfloor j/k\rfloor\eta}) + \xi_j, \quad x_{(j+1)\eta/k} = x_{j\eta/k} + \frac{\eta}{k}v_{(j+1)\eta/k},$$

and $\xi_j \sim N(0, 2\gamma\mu\frac{\eta}{k}I_d)$. Then if $\|x_{(j+1)\eta/k} - x_{\lfloor(j+1)/k\rfloor\eta}\|_2 > r$ (i.e. $\mathcal{E}_r$ no longer holds) replace $(x_{(j+1)\eta/k}, v_{(j+1)\eta/k})$ with $\perp$.

Let $\psi$ denote this update, i.e. $X_{0:j+1} = \psi(X_{0:j})$.

• To construct a sample from $X'_{0:Tk}$, the update (which we denote $\psi'$) is identical to $\psi$ except we use the gradient at $x'_{j\eta/k}$ instead of $x'_{\lfloor j/k\rfloor\eta}$ to compute $v_{(j+1)\eta/k}$.

We remark that unlike in our analysis of the overdamped Langevin dynamics, for finite $k$, $X_{0:Tk}, X'_{0:Tk}$ do *not* actually correspond to the SDE (1) with step size $\eta, \eta/k$. However, we still have the property that the limit of $X_{0:Tk}$ (resp. $X'_{0:Tk}$) as $k$ goes to infinity follows a discretized (resp. continuous) underdamped Langevin dynamics, which is all that is needed for our analysis. Similarly to Theorem 9 we have:

**Theorem 16.** *For any $1$-strongly convex, $L$-smooth function $f$, let $P$ be the distribution of states for discretized underdamped Langevin dynamics with step size $\eta$ and $Q$ be the distribution of states for continuous underdamped Langevin dynamics, both run from any initial distribution on $x_0, v_0$ satisfying appropriate tail bounds, for continuous time $\tau$ that is a multiple of $\eta$. Then for $\alpha > 1$, $\varepsilon > 0$, if $\eta = \tilde{O}(\min\{\frac{1}{L\tau\mu\ln\alpha} \cdot \frac{\varepsilon}{\sqrt{d}}, \frac{\gamma}{\mu L}\})$ we have $D_\alpha(P\|Q) \le \varepsilon$.*

We give here some intuition for why the proof achieves an iteration complexity for underdamped Langevin dynamics with a quadratically improved dependence on $d, \varepsilon$ compared to overdamped Langevin dynamics. The tail bound on the maximum movement within each step of size $\eta$ (and in turn the norm of the discretization error due to the gradient) has a quadratically stronger dependence on $\eta$ in the underdamped case than in the overdamped case. In turn, in underdamped Langevin dynamics the "privacy loss" of hiding this error with Brownian motion also improves quadratically as a function of $\eta$.

## 7 Discussion and Open Questions

Our work raises several interesting questions. While our bounds are for log-smooth and strongly log-concave distributions, it would be interesting to relax these assumptions. The known results for the continuous process in the underdamped case are only for weaker measures, and it is compelling to extend them to Rényi divergence. Our result has a seemingly curious property: the finite time behaviour of the discrete chain is shown to be close in Rényi divergence to the target distribution, yet we do not know if the stationary distribution of the discrete chain satisfies this property. Addressing this gap in our understanding is left to future work. There are several variants of these methods that have been studied (e.g. Metropolis Adjusted Langevin Algorithm, Hamiltonian Monte Carlo, Stochastic Gradient Langevin Dynamics) and extending our techniques to these methods would be interesting. Finally, applying these tools to specific non-convex functions of interest such as the Rayleigh quotient may lead to more practical efficient algorithms for problems such as private PCA [Kapralov and Talwar, 2013].

We note that our bound on iteration complexity for the overdamped Langevin dynamics are proportional to $\tilde{O}(1/\varepsilon^2)$, as opposed to e.g. a $O(1/\varepsilon^{1/2})$ dependence in Mou et al. [2019] for KL-divergence. In many differential privacy applications we would set $\varepsilon$ to be not too small a constant, so this gap may be acceptable from a practical standpoint. Obtaining better dependencies on $\varepsilon$ remains an interesting question. We believe the loss of a $1/\varepsilon^2$ factor in our "unconditioning" argument is unavoidable, and so alternate analyses may be needed to improve this dependence.

## Broader Impact

This work gives faster algorithms for a class of differentially private algorithms. The use of differentially private algorithms has in many cases such as the US Census Bureau, allowed release of useful aggregate statistics while protecting privacy of individual respondents. In some cases, differentially private algorithms may have lower utility compared to those which do not enjoy provable privacy,

which may otherwise be used. Differentially private algorithms give a way to quantify privacy loss and can help decision makers choose an appropriate points on the Pareto curve. Works such as ours will (over time) enable better privacy-utility trade-offs amongst computationally efficient algorithms and thus push the Pareto curve.

## Funding

Arun Ganesh was supported in part by NSF Award CCF-1535989.

## Footnotes

[3]The letter $\varepsilon$ commonly denotes the privacy parameter in DP literature, and the distance to the target distribution in the sampling literature. Since most of the technical part of this work deals with sampling, we will reserve $\varepsilon$ for distance, and will let $\zeta$ denote the privacy parameter.

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
