[Supplementary Material]

# A Preliminaries for Deferred Proofs

## A.1 Rényi Divergence Facts

We state additional facts about Rényi divergences that are needed in our proofs.

**Fact 17** (Monotonicity [van Erven and Harremos, 2014, Theorem 3]). *For any distributions $P, Q$ and $0 \leq \alpha_1 \leq \alpha_2$ we have $D_{\alpha_1}(P||Q) \leq D_{\alpha_2}(P||Q)$.*

**Fact 18** (Post-Processing [van Erven and Harremos, 2014, Theorem 9]). *For any sample spaces $\mathcal{X}, \mathcal{Y}$, distributions $X_1, X_2$ over $\mathcal{X}$, and any function $f : \mathcal{X} \to \mathcal{Y}$ we have $D_{\alpha}(f(X_1)||f(X_2)) \leq D_{\alpha}(X_1||X_2)$.*

**Fact 19** (Gaussian Divergence [van Erven and Harremos, 2014, Example 3]).

$$D_{\alpha}(N(0, \sigma^2 I_d)||N(x, \sigma^2 I_d)) \leq \frac{\alpha \|x\|_2^2}{2\sigma^2}.$$

**Fact 20** (Adaptive Composition Theorem [Mironov, 2017, Proposition 1]). *Let $\mathcal{X}_0, \mathcal{X}_1, \ldots, \mathcal{X}_k$ be arbitrary sample spaces. For each $i \in [k]$, let $\psi_i, \psi_i' : \Delta(\mathcal{X}_{i-1}) \to \Delta(\mathcal{X}_i)$ be maps from distributions over $\mathcal{X}_{i-1}$ to distributions over $\mathcal{X}_i$ such that for any point mass distribution (a distribution whose support contains a single value) $X_{i-1}$ over $\mathcal{X}_{i-1}$, $D_{\alpha}(\psi_i(X_{i-1})||\psi_i'(X_{i-1})) \leq \varepsilon_i$. Then, for $\Psi, \Psi' : \Delta(\mathcal{X}_0) \to \Delta(\mathcal{X}_k)$ defined as $\Psi(\cdot) = \psi_k(\psi_{k-1}(\ldots \psi_1(\cdot) \ldots))$ and $\Psi'(\cdot) = \psi_k'(\psi_{k-1}'(\ldots \psi_1'(\cdot) \ldots))$ we have $D_{\alpha}(\Psi(X_0)||\Psi'(X_0)) \leq \sum_{i=1}^{k} \varepsilon_i$ for any $X_0 \in \Delta(\mathcal{X}_0)$.*

**Fact 21** (Weak Triangle Inequality [Mironov, 2017, Proposition 11]). *For any $\alpha > 1$, $p, q > 1$ satisfying $1/p + 1/q = 1$ and distributions $P, Q, R$ with the same support:*

$$D_{\alpha}(P||R) \leq \frac{\alpha - 1/p}{\alpha - 1} D_{p\alpha}(P||Q) + D_{q(\alpha-1/p)}(Q||R).$$

## A.2 Gaussians and Brownian Motion

We give some standard tail bounds on Gaussians and Brownian motion that will be useful:

**Fact 22** (Univariate Gaussian Tail Bound). *For $X \sim N(0, \sigma^2)$ and any $x \geq 0$, we have*

$$\Pr[X \geq x] = \Pr[X \leq -x] \leq \exp\left(-\frac{x^2}{2\sigma^2}\right).$$

**Fact 23** (Isotropic Multivariate Normal Tail Bound). *For $X \sim N(\mathbf{0}, I_d)$ and any $x \geq 0$, we have*

$$\Pr[\|X\|_2 \geq \sqrt{d} + x] \leq \exp\left(-\frac{x^2}{2}\right).$$

**Fact 24** (Univariate Brownian Motion Tail Bound). *Let $B_t$ be a standard (one-dimensional) Brownian motion. For any $0 \leq a \leq b$, we have:*

$$\Pr\left[\sup_{t \in [a,b]} [B_t - B_a] \geq x\right] = 2 \cdot \Pr[N(0, b-a) \geq x] \leq 2 \exp\left(-\frac{x^2}{2(b-a)}\right)$$

The preceding fact is also known as *the reflection principle*.

**Fact 25** (Multivariate Brownian Motion Tail Bound). *Let $B_t$ be a standard $d$-dimensional Brownian motion. For any $0 \leq a \leq b$, we have:*

$$\Pr\left[\sup_{t \in [a,b]} \|B_t - B_a\|_2 \geq \sqrt{b-a}\left(\sqrt{d} + x\right)\right] \leq 2 \exp(-x^2/4).$$

## A.3 Gradient Descent

For completeness, we recall the contractivity properties of gradient descent.

**Fact 26** (Discrete Gradient Descent Contracts). *Let $f : \mathbb{R}^d \to \mathbb{R}$ be a 1-strongly convex, L-smooth function. Then for $\eta \leq \frac{2}{L+1}$, we have $\|x - \eta\nabla f(x) - x' + \eta\nabla f(x')\|_2 \leq (1 - \frac{\eta L}{L+1})\|x - x'\|_2 \leq (1 - \frac{\eta}{2})\|x - x'\|_2$ for any $x, x' \in \mathbb{R}^d$.*

See e.g. [Hardt et al., 2016, Lemma 3.7] for a proof of this fact.

Since we assume $f$'s global minimum is at 0 (and thus $\nabla f(0) = 0$), as a corollary we have $\|x - \eta\nabla f(x)\|_2 \leq (1 - \eta/2)\|x\|_2$. We also have as a corollary:

**Fact 27** (Continuous Gradient Descent Contracts). *Let $f : \mathbb{R}^d \to \mathbb{R}$ be a 1-strongly convex, L-smooth function. Then for any $x_0, x_0' \in \mathbb{R}^d$ and $x_t, x_t'$ that are solutions to the differential equation $dx_t = -\nabla f(x_t)dt$ we have $\|x_t - x_t'\|_2 \leq e^{-t/2}\|x_0 - x_0'\|_2$.*

*Proof.* This follows by noting that the $x_t$ is the limit as integer $k$ goes to $\infty$ of applying $k$ discrete gradient descent steps to $x_0$ with $\eta = t/k$. So, the contractivity bound we get for $x_t$ is $\|x_t\|_2 \leq \lim_{k\to\infty}(1 - t/2k)^k\|x_0\|_2 = e^{-t/2}\|x_0\|_2$. $\qquad\square$

# B  Deferred Proofs From Section 3

## B.1  Proof of Lemma 5

*Proof.* We prove the bound for $D_\alpha(X_{0:Tk}\|X_{0:Tk}')$, the bound for $D_\alpha(X_{0:Tk}'\|X_{0:Tk})$ follows similarly. Let a tuple $\{x_{i\eta/k}\}_{0\leq i\leq j}$ be *good* if each $x_{i\eta/k}$ satisfies either (i) $\|x_{i\eta/k} - x_{\lfloor i/k\rfloor\eta}\|_2 \leq r$ (i.e., $\mathcal{E}_r$) or (ii) $\{x_{\ell\eta/k}\}_{i\leq\ell\leq j}$ are all $\perp$. We claim that for each $j$, for any point mass distribution $X_{0:j}$ over good $(j + 1)$-tuples:

$$D_\alpha(\psi(X_{0:j}), \psi'(X_{0:j})) \leq \frac{\alpha(\frac{Lr\eta}{k})^2}{2 \cdot \frac{2\eta}{k}}. \tag{3}$$

By Fact 18, we can instead bound the divergence between $\tilde{\psi}(X_{0:j}), \tilde{\psi}'(X_{0:j})$ which are defined equivalently to $\psi, \psi'$ except without the step of replacing the last entry with $\perp$ if $\mathcal{E}_r$ is violated. If $X_{0:j}$ is a point mass on a good tuple containing $\perp$, then $D_\alpha(\tilde{\psi}(X_{0:j})\|\tilde{\psi}'(X_{0:j})) = 0$. For $X_{0:j}$ that is a point mass on a good tuple not containing $\perp$, $D_\alpha(\tilde{\psi}(X_{0:j})\|\tilde{\psi}'(X_{0:j}))$ is just the divergence between the final values of $\tilde{\psi}(X_{0:j}), \tilde{\psi}'(X_{0:j})$. The distance between the final values in $\tilde{\psi}(X_{0:j}), \tilde{\psi}'(X_{0:j})$ prior to the addition of Gaussian noise in $\tilde{\psi}, \tilde{\psi}'$ is the value of $\frac{\eta}{k}\|\nabla f(x_{j\eta/k}) - \nabla f(x_{\lfloor j/k\rfloor\eta})\|_2$ for the single tuple in the support of $X_{0:j}$, which is at most $\frac{Lr\eta}{k}$ by smoothness and because $\mathcal{E}_r$ holds for all good tuples not containing $\perp$. (3) now follows by Fact 19.

Then, $X_{0:Tk}, X_{0:Tk}'$ are arrived at by a composition of $Tk$ applications of $\psi, \psi'$ to the same initial distribution $X_0$. Note that $X_0$ and the distributions arrived at by applying $\psi$ or $\psi'$ any number of times to $X_0$ have support only including good tuples. Then combining Fact 20 (with the sample spaces being good tuples) and (3) we have:

$$D_\alpha(X_{0:Tk}\|X_{0:Tk}') \leq Tk \cdot \frac{\alpha\left(\frac{Lr\eta}{k}\right)^2}{2 \cdot \frac{2\eta}{k}} = \frac{T\alpha L^2 r^2\eta}{4}.$$

$\qquad\square$

# C  Deferred Proofs From Section 4

## C.1  Proof of Lemma 7

*Proof.* We consider the discrete chain first. For each timestep starting at $t$ that is a multiple of $\eta$, using smoothness we have:

$$\max_{t' \in [t,t+\eta)} \|x_{t'} - x_t\|_2 = \max_{t' \in [t,t+\eta)} \| -(t'-t)\nabla f(x_t) + \sqrt{2} \int_t^{t'} dB_s \|_2$$

$$\leq \eta \|\nabla f(x_t)\|_2 + \sqrt{2} \max_{t' \in [t,t+\eta)} \| \int_t^{t'} dB_s \|_2$$

$$\leq \eta L \|x_t\|_2 + \sqrt{2} \max_{t' \in [t,t+\eta)} \| \int_t^{t'} dB_s \|_2.$$

Using the tail bound for multivariate Brownian motion, $\max_{t' \in [t,t+\eta)} \| \int_t^{t'} dB_s \|_2$ is at most $\frac{c}{2\sqrt{2}} \left( \sqrt{d} + \sqrt{\ln(T/\delta)} \right) \sqrt{\eta}$ with probability at least $1 - \frac{\delta}{2T}$ for each timestep. So it suffices to show that with probability at least $1 - \frac{\delta}{2}$, for all $0 \leq t < T\eta$ that are multiples of $\eta$, $\|x_t\|_2 \leq \frac{c}{2\sqrt{\eta}} \left( \sqrt{d} + \sqrt{\ln(T/\delta)} \right)$. From (2), with probability $1 - \frac{\delta}{T+1}$, $\|x_0\|_2 \leq \frac{c}{2\sqrt{\eta}} \left( \sqrt{d} + \sqrt{\ln(T/\delta)} \right)$. We will show that if $\|x_t\|_2 \leq \frac{c}{2\sqrt{\eta}} \left( \sqrt{d} + \sqrt{\ln(T/\delta)} \right)$ then with probability $1 - \frac{\delta}{T+1}$ we have $\|x_{t+\eta}\|_2 \leq \frac{c}{2\sqrt{\eta}} \left( \sqrt{d} + \sqrt{\ln(T/\delta)} \right)$, completing the proof for the discrete case by a union bound. This follows because by Fact 26 the gradient descent step is $(1 - \eta/2)$-Lipschitz for the range of $\eta$ we consider. This gives that after the gradient descent step but before adding Gaussian noise, $x_{t+\eta}$ has norm at most $(1 - \eta/2)\|x_t\|_2 \leq (1 - \eta/2)\frac{c}{2\sqrt{\eta}} \left( \sqrt{d} + \sqrt{\ln(T/\delta)} \right)$. Then, $\|x_{t+\eta}\|_2 > \frac{c}{2\sqrt{\eta}} \left( \sqrt{d} + \sqrt{\ln(T/\delta)} \right)$ only if $\sqrt{2}\| \int_t^{t+\eta} dB_s \|_2$ is larger than $c\sqrt{\eta} \left( \sqrt{d} + \sqrt{\ln(T/\delta)} \right)$, which happens with probability at most $\frac{\delta}{T+1}$ by the multivariate Gaussian tail bound.

We now consider the continuous chain. For all $t$ that are multiples of $\eta$:

$$\max_{u \in [t,t+\eta)} \|x'_u - x'_t\|_2 = \max_{u \in [t,t+\eta)} \| \int_t^u -\nabla f(x'_s) ds + \sqrt{2} dB_s \|_2$$

$$\leq \eta L \max_{u \in [t,t+\eta)} \|x'_u\|_2 + \max_{u \in [t,t+\eta)} \| \sqrt{2} \int_t^u dB_s \|_2.$$

As with the discrete chain, the multivariate Brownian motion tail bound gives that

$$\max_{u \in [t,t+\eta)} \| \sqrt{2} \int_t^u dB_s \|_2 \leq \frac{c}{2} \left( \sqrt{d} + \sqrt{\ln(T/\delta)} \right) \sqrt{\eta},$$

with probability at least $1 - \frac{\delta}{2T}$. So it suffices to show that at all times between 0 and $T\eta$, $\|x'_u\|_2 \leq \frac{c}{2\sqrt{\eta}} \left( \sqrt{d} + \sqrt{\ln(T/\delta)} \right)$ with probability at least $1 - \frac{\delta}{2}$. We first claim that with probability at least $1 - \frac{\delta}{4}$, for all $t$ that are multiples of $\eta$, $\|x'_t\|_2 \leq \frac{c}{4\sqrt{\eta}} \left( \sqrt{d} + \sqrt{\ln(T/\delta)} \right)$. This is true for $x'_0$ with probability at least $1 - \frac{\delta}{4(T+1)}$ by (2). By contractivity of continuous gradient descent, $x'_{t+\eta}$ is equal to $Ax'_t + \sqrt{2} \int_t^{t+\eta} A'_s dB_s$ for some $A$ which has eigenvalues in $[-e^{-\eta/2}, e^{-\eta/2}]$ and a set of matrices $\{A'_s | s \in [0,\eta]\}$ with eigenvalues in $[-e^{-(\eta-s)/2}, e^{-(\eta-s)/2}]$[4]. Then conditioning on the claim holding for $x'_t$, $\|x'_{t+\eta}\|_2$ exceeds $\frac{c}{4\sqrt{\eta}} \left( \sqrt{d} + \sqrt{\ln(T/\delta)} \right)$ only if the norm of $\sqrt{2} \int_t^{t+\eta} A'_s dB_s$ exceeds $\frac{c(1-e^{-\eta/2})}{4\sqrt{\eta}} \left( \sqrt{d} + \sqrt{\ln(T/\delta)} \right) \geq \frac{c(1-e^{-.5})\sqrt{\eta}}{4} \left( \sqrt{d} + \sqrt{\ln(T/\delta)} \right)$. Since Brownian motion is rotationally symmetric, and all $A'_s$ have eigenvalues in $[-1, 1]$, this occurs with probability upper bounded by the probability $\sqrt{2} \int_t^{t+\eta} dB_s$ exceeds this bound, which is at most $\frac{\delta}{4(T+1)}$ by the

Brownian motion tail bound. The claim follows by taking a union bound over all $t$ that are multiples of $\eta$.

Then, conditioning on the event in the claim, for each corresponding interval $[t, t + \eta)$ since gradient descent contracts we have

$$\max_{u \in [t, t+\eta)} \|x'_u\|_2 \leq \|x'_t\|_2 + \max_{u \in [t, t+\eta)} \|\sqrt{2} \int_t^u \mathrm{d}B_s\|_2$$

$$\leq \frac{c}{4\sqrt{\eta}} \left(\sqrt{d} + \sqrt{\ln(T/\delta)}\right) + \max_{u \in [t, t+\eta)} \|\sqrt{2} \int_t^u \mathrm{d}B_s\|_2.$$

We conclude by using the multivariate Brownian motion tail bound to observe that

$$\max_{u \in [t, t+\eta)} \|\sqrt{2} \int_t^u \mathrm{d}B_s\|_2 \leq \frac{c}{4\sqrt{\eta}} \left(\sqrt{d} + \sqrt{\ln(T/\delta)}\right),$$

with probability at least $1 - \frac{\delta}{4T}$, and then taking a union bound over all intervals. $\square$

## C.2  Proof of Lemma 8

*Proof.* Let $z$ be an arbitrary parameter, $\eta : [z, \infty) \to (0, 1/2)$ be an arbitrary map, and $\mathcal{E}_\delta$ be the event specified in the lemma statement for $\delta \in (0, 1)$. Using the definition of expectation, we have:

$$\mathbb{E}[Y] = \int_0^\infty \Pr[Y \geq y]\mathrm{d}y$$

$$\leq \int_0^z 1 \, \mathrm{d}y + \int_z^\infty \Pr[Y \geq y]\mathrm{d}y$$

$$\leq z + \int_z^\infty \eta(y) + (1 - \eta(y))\Pr[Y \geq y | \mathcal{E}_{\eta(y)}]\mathrm{d}y$$

$$\leq z + \int_z^\infty \eta(y) + \Pr[Y \geq y | \mathcal{E}_{\eta(y)}]\mathrm{d}y$$

$$= z + \int_z^\infty \eta(y) + \Pr[Y^\theta \geq y^\theta | \mathcal{E}_{\eta(y)}]\mathrm{d}y$$

$$\leq z + \int_z^\infty \eta(y) + \frac{\mathbb{E}[Y^\theta | \mathcal{E}_{\eta(y)}]}{y^\theta}\mathrm{d}y$$

$$\leq z + \int_z^\infty \eta(y) + \frac{\beta}{\eta(y)^\gamma y^\theta}\mathrm{d}y.$$

We now choose $\eta(y) = \left(\frac{\gamma\beta}{y^\theta}\right)^{\frac{1}{1+\gamma}}$ to minimize the value of the expression in the integral. We will eventually choose $z$ such that $0 < \eta(y) < 1/2$ for all $y \geq z$ as is required of $\eta$. Plugging in this choice of $\eta$ gives the upper bound:

$$\mathbb{E}[Y] \leq z + \beta^{\frac{1}{1+\gamma}}(\gamma^{\frac{1}{1+\gamma}} + \gamma^{-\frac{\gamma}{1+\gamma}}) \int_z^\infty y^{-\frac{\theta}{1+\gamma}}\mathrm{d}y$$

$$= z + \beta^{\frac{1}{1+\gamma}}(\gamma^{\frac{1}{1+\gamma}} + \gamma^{-\frac{\gamma}{1+\gamma}}) \left(\frac{1}{\frac{\theta}{1+\gamma} - 1}\right) \left[y^{1 - \frac{\theta}{1+\gamma}}\right]_\infty^z$$

$$= z + \beta^{\frac{1}{1+\gamma}}(\gamma^{\frac{1}{1+\gamma}} + \gamma^{-\frac{\gamma}{1+\gamma}}) \left(\frac{1}{\frac{\theta}{1+\gamma} - 1}\right) z^{1 - \frac{\theta}{1+\gamma}}.$$

We finish by choosing $z = \beta^{\frac{1}{\theta}} \left( \gamma^{\frac{1}{1+\gamma}} + \gamma^{-\frac{\gamma}{1+\gamma}} \right)^{\frac{1+\gamma}{\theta}}$. This gives the upper bound on $\mathbb{E}[Y]$ in the lemma statement. We also verify that $\eta(y)$ is a map to $(0, 1/2)$: $\eta(y) \propto y^{-\frac{\theta}{1+\gamma}}$, giving that $\eta(y) > 0$. For all $y \geq z$, since $\gamma < 1$ we have $\eta(y) \leq \eta(z) = \frac{\gamma}{\gamma+1} < 1/2$. $\qquad\square$

## C.3 Proof of Theorem 9

*Proof.* We bound $D_\alpha(P||Q)$, the bound on $D_\alpha(Q||P)$ follows similarly. We first need the following corollary of Lemma 5, which follows from that lemma by taking the limit as $k$ goes to infinity and applying Fact 18:

**Corollary 28.** *For any $L$-smooth $f$ and $\eta > 0$, and any initial distribution $X_0$ let $X_t$ be the distribution over positions $x_t$ arrived at by running the discretized underdamped Langevin dynamics with step size $\eta$ on $f$ from $X_0$ for continuous time $t$, except that $X_t = \perp$ if $\mathcal{E}_r$ does not hold at time $t$ for this chain. Let $X'_t$ be the same but for the continuous underdamped Langevin dynamics. Then for any integer $T \geq 0$:*

$$D_\alpha(X_{T\eta}||X'_{T\eta}), D_\alpha(X'_{T\eta}||X_{T\eta}) \leq \frac{T\alpha L^2 r^2 \eta}{4}.$$

For arbitrary $\delta_1, \delta_2$, plugging in $r = cL(\sqrt{d} + \sqrt{\ln(T/\delta_1)} + \sqrt{\ln(T/\delta_2)})\sqrt{\eta}$ into Corollary 28 (where $c$ is the constant specified in Lemma 7) and using the definition $T = \tau/\eta$ we get that

$$D_{\alpha'}(X_{T\eta}||X'_{T\eta}) \leq \frac{3\tau\alpha'L^4c^2(d + \ln(\frac{\tau}{\eta\delta_1}) + \ln(\frac{\tau}{\eta\delta_2}))\eta}{4}$$

for all $k \in \mathbb{Z}^+$ and $X_{T\eta}, X'_{T\eta}$ as defined in Corollary 28. Using the definition of Rényi divergence, this gives:

$$\int_{\mathbb{R}^d} \frac{X_{T\eta}(x)^{\alpha'}}{X'_{T\eta}(x)^{\alpha'-1}} \mathrm{d}x \leq \int_{\mathbb{R}^d} \frac{X_{T\eta}(x)^{\alpha'}}{X'_{T\eta}(x)^{\alpha'-1}} \mathrm{d}x + \frac{\Pr_{x\sim X_{T\eta}}[x = \perp]^{\alpha'}}{\Pr_{x\sim X'_{T\eta}}[x = \perp]^{\alpha'-1}} \leq \frac{c_1(\alpha')}{\delta_1^{c_2(\alpha')}\delta_2^{c_3(\alpha')}},$$

where:

$$c_1(\alpha') = \exp\left( \frac{3\tau\alpha'(\alpha'-1)L^4c^2(d + 2\ln(\frac{\tau}{\eta}))\eta}{4} \right),$$

$$c_2(\alpha') = c_3(\alpha') = \frac{3\tau\alpha'(\alpha'-1)L^4c^2\eta}{4}.$$

**Removing the conditioning on the continuous chain:** Let $\mathcal{E}_{\delta_1}$ denote the (at least probability $1-\delta_1$) event that the conditions in Lemma 7 are satisfied for the discrete chain and $\mathcal{E}_{\delta_2}$ denote the (at least probability $1 - \delta_2$) event that the conditions in Lemma 7 are satisfied for the continuous chain. By Lemma 7, we have $Q(x) \geq X'_{T\eta}(x), Q(x|\mathcal{E}_{\delta_2}) \leq \frac{1}{1-\delta_2}X'_{T\eta}(x)$. Then for $\delta_2 < 1/2$:

$$\mathbb{E}_{x\sim Q}\left[ \frac{X_{T\eta}(x)^{\alpha'}}{Q(x)^{\alpha'}} \middle| \mathcal{E}_{\delta_2} \right] = \int_{\mathbb{R}^d} Q(x|\mathcal{E}_{\delta_2}) \frac{X_{T\eta}(x)^{\alpha'}}{Q(x)^{\alpha'}} \mathrm{d}x$$

$$\leq \frac{1}{1-\delta_2} \int_{\mathbb{R}^d} \frac{X_{T\eta}(x)^{\alpha'}}{X'_{T\eta}(x)^{\alpha'-1}} \mathrm{d}x$$

$$\leq \frac{2 \cdot c_1(\alpha')}{\delta_1^{c_2(\alpha')}\delta_2^{c_3(\alpha')}}.$$

This statement holds independent of $\delta_2$. We will eventually choose $\alpha'$ such that for the choice of $\eta$ specified in the lemma statement, $c_1(\alpha') < 2, c_3(\alpha') < 1$. Then applying Lemma 8 with $Y = \frac{X_{T\eta}(x)^{\alpha'/2}}{Q(x)^{\alpha'/2}}$ $\theta = 2, \beta = \frac{2c_1(\alpha')}{\delta_1^{c_2(\alpha')}}, \gamma = c_3(\alpha')$, we get:

$$\mathbb{E}_{x \sim Q}\left[\frac{X_{T\eta}(x)^{\alpha'/2}}{Q(x)^{\alpha'/2}}\right] \leq \frac{8}{\delta_1^{c_2(\alpha')/2}}.$$

**Removing the conditioning on the discrete chain:** We now turn to removing the conditioning on $\mathcal{E}_{\delta_1}$. Here we need to be a bit more careful since unlike with $X'_{T\eta}(x)$, $X_{T\eta}(x)$ is in the numerator and so the inequality $X_{T\eta}(x) \leq P(x)$ is facing the wrong way. Since $P, Q$ have the same support, we note that:

$$
\begin{aligned}
\mathbb{E}_{x \sim Q}\left[\frac{X_{T\eta}(x)^{\alpha'/2}}{Q(x)^{\alpha'/2}}\right] &= \mathbb{E}_{x \sim P}\left[\frac{X_{T\eta}(x)^{\alpha'/2-1}}{Q(x)^{\alpha'/2-1}}\right] \\
&\overset{(\star)}{\geq} \frac{\alpha'}{2}\mathbb{E}_{x \sim P, y \sim Unif(0,P(x))}\left[\frac{y^{\alpha'/2-1}}{Q(x)^{\alpha'/2-1}} \cdot \mathbb{I}\left[y \leq X_{T\eta}(x)\right]\right] \\
&= \frac{\alpha'}{2}\mathbb{E}_{x \sim P, y \sim Unif(0,P(x))}\left[\frac{y^{\alpha'/2-1}}{Q(x)^{\alpha'/2-1}}\bigg| y \leq X_{T\eta}(x)\right] \\
&\qquad \cdot \Pr_{x \sim P, y \sim Unif(0,P(x))}\left[y \leq X_{T\eta}(x)\right] \\
&\geq \frac{\alpha'}{2}\mathbb{E}_{x \sim P, y \sim Unif(0,P(x))}\left[\frac{y^{\alpha'/2-1}}{Q(x)^{\alpha'/2-1}}\bigg| \mathcal{E}_{\delta_1}\right] \cdot (1 - \delta_1).
\end{aligned}
$$

$(\star)$ follows as for any given any $x$, we have:

$$
\begin{aligned}
X_{T\eta}(x)^{\alpha'/2-1} &= \frac{1}{X_{T\eta}(x)}X_{T\eta}(x)^{\alpha'/2} \\
&= \int_0^{X_{T\eta}(x)} \frac{1}{X_{T\eta}(x)}\frac{\alpha'}{2}y^{\alpha'/2-1}\mathrm{d}y \\
&\geq \int_0^{X_{T\eta}(x)} \frac{1}{P(x)}\frac{\alpha'}{2}y^{\alpha'/2-1}\mathrm{d}y \\
&= \int_0^{P(x)} \frac{1}{P(x)}\frac{\alpha'}{2}y^{\alpha'/2-1} \cdot \mathbb{I}\left[y \leq X_{T\eta}(x)\right]\mathrm{d}y \\
&= \frac{\alpha'}{2}\mathbb{E}_{y \sim Unif(0,P(x))}\left[y^{\alpha'/2-1} \cdot \mathbb{I}\left[y \leq X_{T\eta}(x)\right]\right].
\end{aligned}
$$

In turn, for all $\delta_1 < 1/2$, we have

$$\mathbb{E}_{x \sim P, y \sim Unif(0,P(x))}\left[\frac{y^{\alpha'/2-1}}{Q(x)^{\alpha'/2-1}}\bigg| \mathcal{E}_{\delta_1}\right] \leq \frac{32}{\alpha'\delta_1^{c_2(\alpha')/2}}.$$

If $c_2(\alpha')/2 < 1/2$ (which is equivalent to $c_2(\alpha') = c_3(\alpha') < 1$), by applying Lemma 8 for $\theta = 2$ with $X = \frac{y^{\alpha'/4-1/2}}{Q(x)^{\alpha'/4-1/2}}, \beta = \frac{32}{\alpha'}, \gamma = c_2(\alpha')/2$ we get:

$$\mathbb{E}_{x \sim P, y \sim Unif(0,P(x))}\left[\frac{y^{\alpha'/4-1/2}}{Q(x)^{\alpha'/4-1/2}}\right] \leq \frac{19}{\sqrt{\alpha'}} \implies$$

$$\mathbb{E}_{x \sim Q}\left[\frac{P(x)^{\alpha'/4+1/2}}{Q(x)^{\alpha'/4+1/2}}\right] = \left(\frac{\alpha'}{4} + \frac{1}{2}\right) \mathbb{E}_{x \sim P, y \sim Unif(0, P(x))}\left[\frac{y^{\alpha'/4-1/2}}{Q(x)^{\alpha'/4-1/2}}\right]$$

$$\leq \frac{19(\alpha'/4 + 1/2)}{\sqrt{\alpha'}}$$

$$\leq 15\sqrt{\alpha'}.$$

**From moderate $\alpha'$-Rényi divergence to small $\alpha$-Rényi divergence:** If $\varepsilon \geq \frac{3 \ln \alpha}{\alpha - 1}$, without loss of generality we can assume e.g. $\alpha \geq 4$ (by monotonocity of Rényi divergences, if $\alpha < 4$ it suffices to bound the 4-Rényi divergence instead of the $\alpha$-Rényi divergence at the loss of a constant in the bound for $\eta$). Then for $\alpha' = 4\alpha - 2$ this inequality lets us conclude the lemma holds. Otherwise, for $1 < \kappa < \alpha'/4 + 1/2$, for $\alpha = \frac{\alpha'/4+1/2}{\kappa}$, by Jensen's inequality we get:

$$\frac{1}{\alpha - 1} \ln \mathbb{E}_{x \sim Q}\left[\frac{P(x)^\alpha}{Q(x)^\alpha}\right] \leq \frac{1}{\alpha - 1} \ln \left(\mathbb{E}_{x \sim Q}\left[\frac{P(x)^{\alpha\kappa}}{Q(x)^{\alpha\kappa}}\right]^{1/\kappa}\right) \leq \frac{\ln 15 + \frac{1}{2} \ln \alpha + \frac{1}{2} \ln \kappa}{(\alpha - 1)\kappa}.$$

Choosing $\kappa = \frac{3 \ln \alpha \cdot \ln 1/\varepsilon}{(\alpha-1)\varepsilon}$ then gives $D_\alpha(P||Q) \leq \varepsilon$ as desired (note that for $\varepsilon < \frac{3 \ln \alpha}{\alpha-1}$ we have $\kappa > 1$ as is required). Now, we just need to verify that $c_1(\alpha') < 2, c_2(\alpha') = c_3(\alpha') < 1$ holds for $\alpha' = \frac{12\alpha \ln \alpha \cdot \ln 1/\varepsilon}{(\alpha-1)\varepsilon} - 2$. Since $c_2(\alpha') = c_3(\alpha') < \ln(c_1(\alpha'))/d$, it just suffices to show $c_1(\alpha') < 2$. This holds if:

$$\frac{3\tau\alpha'(\alpha'-1)L^4 c^2(d + 2\ln(\frac{\tau}{\eta}))\eta}{4} < \ln 2,$$

which is given by choosing $\eta = \tilde{O}(\frac{1}{\tau L^4 \ln^2 \alpha} \cdot \frac{\varepsilon^2}{d})$ with a sufficiently small constant hidden in $\tilde{O}$. $\square$

### C.4 Proof of Lemma 10

*Proof.* This follows from Lemma 4 in Vempala and Wibisono [2019], which gives the bound $D_\alpha(N(0, \frac{1}{L}I_d)||R) \leq f(\mathbf{0}) + \frac{d}{2} \ln \frac{L}{2\pi}$. We then note that the 1-strongly convex, $L$-smooth $f$ with the maximum $f(\mathbf{0})$ is given when $R$ is $N(0, I_d)$, which has density $R(x) = e^{-\left(\frac{d}{2} \ln(2\pi) + \frac{1}{2} x^\top x\right)}$. $\square$

### C.5 Proof of Theorem 6

*Proof.* We will prove the bound for $\alpha \geq 3/2$ - the bound for $1 \leq \alpha < 3/2$ follows by just applying monotonicity to the bound for $\alpha = 3/2$, at the loss of a multiplicative constant on $\tau, \eta$, and the iteration complexity.

Let $R$ be the distribution arrived at by running continuous overdamped Langevin dynamics using $f$ for time $\tau$ from initial distribution $N(0, \frac{1}{L}I_d)$. $N(0, \frac{1}{L}I_d)$ satisfies (2), so from Theorem 9 we have $D_{2\alpha}(P||Q) \leq \varepsilon/3$. From Lemmas 10 and 11 we have $D_{2\alpha}(Q||R) \leq \varepsilon/3$. Then, we use weak triangle inequality of Rényi divergence with $p, q = 2$ to conclude that $D_\alpha(P||R) \leq \varepsilon$. $\square$

### C.6 Proof of Theorem 12

We have the following radius tail bound:

**Lemma 29.** *For all $\eta \leq 1$ and any $B$-Lipschitz, $L$-smooth $f$, let $x_t$ be the random variable given by running the discretized overdamped Langevin dynamics starting from an arbitrary initial distribution for continuous time $t$. Then with probability $1 - \delta$ over $\{x_t : t \in [0, T\eta]\}$, for all $t \leq T\eta$ and for a sufficiently large constant $c$:*

$$\|x_t - x_{\lfloor t/\eta \rfloor \eta}\|_2 \leq c(B + \sqrt{d} + \sqrt{\ln(T/\delta)})\sqrt{\eta}.$$

*Similarly, if $x_t'$ is the random variable given by running continuous overdamped Langevin dynamics starting from an arbitrary initial distribution for time $t$, with probability $1 - \delta$ over $x_t'$ for all $t \leq T\eta$:*

$$\|x'_t - x'_{\lfloor t/\eta \rfloor \eta}\|_2 \le c(B + \sqrt{d} + \sqrt{\ln(T/\delta)})\sqrt{\eta}.$$

*Proof.* By $B$-Lipschitzness of $f$, the movement in any interval of length $\eta$ due to the gradient step in both the discrete and continuous case is at most $2B\eta$. By the multivariate Brownian motion tail bound, in both the discrete and continuous cases the maximum movement due to the addition of Gaussian noise is at most $c(\sqrt{d} + \sqrt{\ln(T/\delta)})\sqrt{\eta}$ with probability at least $1 - \frac{\delta}{T}$ in each interval of length $\eta$, and then the lemma follows by a union bound and triangle inequality. $\square$

Now Theorem 12 follows identically to Theorem 9, except using Lemma 34 instead of Lemma 7

# D  Deferred Proofs From Section 5

## D.1  Proof of Lemma 13

To prove Lemma 13, we modify the proofs of Lemma 4 and 5 of Vempala and Wibisono [2019]. To describe the modifications, we reintroduce the following definitions from that paper:

**Definition 30.** *We say that a distribution $Q$ has LSI constant $\kappa$ if for all smooth functions $g : \mathbb{R}^n \to \mathbb{R}$ for which $\mathbb{E}_{x \sim Q}[g(x)^2] < \infty$:*

$$\mathbb{E}_{x \sim Q}\left[g(x)^2 \log\left(g(x)^2\right)\right] - \mathbb{E}_{x \sim Q}\left[g(x)^2\right] \log\left(\mathbb{E}_{x \sim Q}\left[g(x)^2\right]\right) \le \frac{2}{\kappa}\mathbb{E}_{x \sim Q}\left[\|\nabla g(x)\|^2\right].$$

**Definition 31.** *We define for $\alpha \ne 0, 1$:*

$$F_\alpha(Q\|R) = \mathbb{E}_{x \sim R}\left[\frac{Q(x)^\alpha}{R(x)^\alpha}\right],$$

$$G_\alpha(Q\|R) = \mathbb{E}_{x \sim R}\left[\frac{Q(x)^\alpha}{R(x)^\alpha}\|\nabla \log \frac{Q(x)}{R(x)}\|_2^2\right] = \frac{4}{\alpha^2}\mathbb{E}_{x \sim R}\left[\|\nabla \left(\frac{Q(x)}{R(x)}\right)^{\alpha/2}\|_2^2\right].$$

*For $\alpha = 0, 1$ these quantities are defined as their limit as $\alpha$ goes to $0, 1$ respectively.*

Unlike Vempala and Wibisono [2019], we extend this definition to negative values of $\alpha$, which allows us to swap the arguments $Q, R$:

**Fact 32.** $F_{1-\alpha}(Q\|R) = F_\alpha(R\|Q), G_{1-\alpha}(Q\|R) = G_\alpha(R\|Q)$. *We also recall that $D_{1-\alpha}(Q\|R) = \frac{1-\alpha}{\alpha}D_\alpha(R\|Q)$.*

*Proof of Lemma 13.* Bakry and Émery [1985] shows that since the initial distribution satisfies that $-\log Q_0$ is 1-strongly convex, $Q_0$ has LSI constant 1. Consider instead running the discrete over-damped Langevin dynamics with step size $\eta$ starting with $Q_0$. In one step, we apply a gradient descent step that is $(1 - \eta/2)$-Lipschitz (see e.g. [Hardt et al., 2016, Lemma 3.7]), and then add Gaussian noise $N(0, 2\eta I_d)$. Lemma 16 in Vempala and Wibisono [2019] shows that applying a $(1-\eta/2)$-Lipschitz map to a distribution with LSI constant $c$ results in a distribution with LSI constant at least $c/(1-\eta/2)^2$. Adding Gaussian noise $N(0, 2\eta I_d)$ to a distribution with LSI constant $c$ results in a distribution with LSI constant at least $\frac{1}{1/c+2\eta}$ (see e.g. [Wang and Wang, 2016, Proposition 1.1]). Putting it together, we get that after one step of the discrete dynamics, the LSI constant of the distribution goes from $c$ to at least:

$$\frac{1}{\frac{(1-\eta/2)^2}{c} + 2\eta} = \frac{c}{1 - (1 - 2c)\eta + \eta^2/4}.$$

Then, we have that $1 - (1 - 2c)\eta + \eta^2/4 \le 1$, i.e. the LSI constant does not decrease after one step, as long as $\eta \le 4(1 - 2c)$. Taking the limit as $\eta$ goes to 0, we conclude that $Q_t$'s LSI constant can never decrease past 1/2, i.e. $Q_t$ has LSI constant at least 1/2 for all $t \ge 0$.

Now, since $Q_t$ has LSI constant at least 1/2, we can repeat the proof of Lemma 5 in Vempala and Wibisono [2019] with the distributions swapped to show that $\frac{G_\alpha(R||Q_t)}{F_\alpha(R||Q_t)} \geq \frac{1}{\alpha^2} D_\alpha(R||Q_t)$. Applying Fact 32 to the proof of Lemma 6 in Vempala and Wibisono [2019], we can show that $\frac{\mathrm{d}}{\mathrm{d}t} D_\alpha(R||Q_t) = -\alpha \frac{G_\alpha(R||Q_t)}{F_\alpha(R||Q_t)}$. Combining these two inequalities and integrating gives the lemma. $\qquad\square$

### D.2   Proof of Lemma 15

The proof of Lemma 15 follows similarly to that of Lemma 10.

*Proof of Lemma 15.* Since $f$ is 1-strongly convex and $L$-smooth, we have:

$$f(\mathbf{0}) + \frac{1}{2}\|x\|_2^2 \leq f(x) \leq f(\mathbf{0}) + \frac{L}{2}\|x\|_2^2.$$

Then:

$$
\begin{aligned}
\exp((\alpha-1)D_\alpha(R||Q_0)) &= \int_{\mathbb{R}^d} \frac{R(x)^\alpha}{Q_0(x)^{\alpha-1}}\mathrm{d}x \\
&= (2\pi)^{d(\alpha-1)/2} \int_{\mathbb{R}^d} \exp\left(-\alpha f(x) + \frac{\alpha-1}{2}\|x\|_2^2\right)\mathrm{d}x \\
&\leq \frac{(2\pi)^{d(\alpha-1)/2}}{e^{\alpha f(\mathbf{0})}} \int_{\mathbb{R}^d} \exp\left(-\frac{1}{2}\|x\|_2^2\right)\mathrm{d}x \\
&= \frac{(2\pi)^{d\alpha/2}}{e^{\alpha f(\mathbf{0})}}.
\end{aligned}
$$

Taking logs and using that the $L$-smooth $f$ that minimizes $f(\mathbf{0})$ is $N(0, \frac{1}{L}I_d)$ with density $\exp(-\frac{d}{2}\log(2\pi/L) - L\|x\|_2^2)$:

$$D_\alpha(R||Q_0) \leq \frac{\alpha}{\alpha-1} \cdot \left(\frac{d}{2}\log 2\pi - f(\mathbf{0})\right) \leq \frac{\alpha}{\alpha-1} \cdot \frac{d}{2}\log L.$$

For $\alpha \geq 2$, the above bound is thus at most $d\log L$ as desired, and for $1 \leq \alpha \leq 2$ we can just use monotonicity of Rényi divergences to bound $D_\alpha(R||Q_0)$ by $D_2(R||Q_0)$.

Similarly:

$$
\begin{aligned}
\exp((1/L)D_{1+1/L}(Q_0||R)) &= \int_{\mathbb{R}^d} \frac{Q_0(x)^{1+1/L}}{R(x)^{1/L}}\mathrm{d}x \\
&= (2\pi)^{-d(1+1/L)/2} \int_{\mathbb{R}^d} \exp\left(-\frac{1+1/L}{2}\|x\|_2^2 + f(x)/L\right)\mathrm{d}x \\
&\leq \frac{e^{f(\mathbf{0})/L}}{(2\pi)^{d(1+1/L)/2}} \int_{\mathbb{R}^d} \exp\left(-\frac{1}{2L}\|x\|_2^2\right)\mathrm{d}x \\
&= \frac{e^{f(\mathbf{0})/L}L^{d/2}}{(2\pi)^{d/2L}}.
\end{aligned}
$$

Taking logs, and using that the 1-strongly convex $f$ that maximizes $f(\mathbf{0})$ is $N(0, I_d)$ with density $\exp(-\frac{d}{2}\log(2\pi) - L\|x\|_2^2)$:

$$D_{1+1/L}(Q_0||R) \leq L\left[f(\mathbf{0})/L + \frac{d}{2}\log L - \frac{d}{2L}\log(2\pi)\right] \leq \frac{dL\log L}{2}.$$

$\qquad\square$

### D.3 Proof of Theorem 1

*Proof of Theorem 1.* Let $Q_t$ be the distribution of the continuous overdamped Langevin dynamics using $f$ run from initial distribution $N(0, I_d)$ for time $t$. Assume without loss of generality that $\alpha \geq 2$. If $\tau$ is at least a sufficiently large constant times $\alpha \ln \frac{d \ln L}{\epsilon}$, Lemma 15 and Lemma 13 give that $D_{2\alpha}(R||Q_\tau) \leq \epsilon/3$. Theorem 9 gives that $D_{2\alpha}(Q_\tau||P) \leq \epsilon/3$. Fact 21 with $p, q = 2$ gives that $D_\alpha(R||P) \leq \epsilon$.

Similarly, Lemma 14 and Lemma 15 give that at time $t = \frac{1}{2} \log((2\alpha - 1)L)$, $D_{2\alpha}(Q_t||R) \leq d \log L$. Then Lemma 11 gives that, $D_{2\alpha}(Q_\tau||R) \leq \epsilon/3$. Theorem 9 gives that $D_{2\alpha}(P||Q_\tau) \leq \epsilon/3$. Fact 21 with $p, q = 2$ again gives that $D_\alpha(P||R) \leq \epsilon$. $\qquad\square$

## E Deferred Proofs From Section 6

### E.1 Proof of Theorem 16

*Proof.* The proof follows similarly to that of Theorem 9. Similarly to the overdamped Langevin dynamics we have:

**Lemma 33.** *For any $L$-smooth $f$ and $X_{0:Tk}, X'_{0:Tk}$ as defined in Section 6, we have:*

$$D_\alpha(X_{0:Tk}||X'_{0:Tk}), D_\alpha(X'_{0:Tk}||X_{0:Tk}) \leq \frac{T\alpha L^2 r^2 \eta}{4} \cdot \frac{\mu}{\gamma}.$$

The proof follows almost exactly as did the proof of Lemma 5: we note that the updates to position are deterministic, and so by Fact 18 we just need to control the divergence between velocities, which can be done using the same analysis as in Lemma 5. The multiplicative factor of $\mu/\gamma$ appears because the ratio of the Gaussian's standard deviation to the gradient step's multiplier is $\sqrt{\gamma/\mu}$ times what it was in the overdamped Langevin dynamics. Next, similar to Lemma 7, we have the following tail bound on $r$:

**Lemma 34.** *Fix any $\gamma \geq 2$, and define*

$$v_{\max} := c\sqrt{\gamma\mu}\left(\sqrt{\tau d} + \sqrt{\ln(1/\delta)}\right).$$

*Fix any $\eta \leq \frac{\gamma}{\mu L}$, and any distribution over $x_0, v_0$ satisfying that*

$$\Pr\left[\mu f(x_0) + \frac{\|v_0\|_2^2}{2} \leq \frac{1}{2}v_{\max}^2\right] \geq 1 - \delta, \tag{4}$$

*let $x_t, v_t$ be the random variable given by running the discretized underdamped Langevin dynamics starting from $x_0, v_0$ drawn from this distribution for time $t$. Then with probability $1 - \delta$ over $\{(x_t, v_t) : t \in [0, \tau]\}$, for all $t \leq \tau$ that are multiples of $\eta$ and for a sufficiently large constant $c$:*

$$\|x_{t+\eta} - x_t\|_2 \leq v_{\max}\eta.$$

*Similarly, if $x_t$ is the random variable given by running continuous underdamped Langevin dynamics starting from $x_0, v_0$ drawn from this distribution for time $t$, with probability $1 - \delta$ over $\{(x'_t, v'_t) : t \in [0, \tau]\}$ for all $t \leq \tau$:*

$$\|x_t - x_{\lfloor t/\eta \rfloor \eta}\|_2 \leq v_{\max}\eta.$$

We give the proof in the following subsection. We note that the correct tail bound likely has a logarithmic dependence on $\tau$ and not a polynomial one. However, based on similar convergence bounds (e.g. Vempala and Wibisono [2019], Ma et al. [2019]), we conjecture that the time $\tau$ needed for continuous underdamped Langevin dynamics to converge in Rényi divergence has a logarithmic dependence on $d, 1/\varepsilon$. So, improving the dependence on $\tau$ in this tail bound will likely not improve

the final iteration complexity's dependence on $d, 1/\varepsilon$ by more than logarithmic factors. In addition, settling for a polynomial dependence on $\tau$ makes the proof rather straightforward.

Finally, from Lemma 33, plugging in the tail bound of Lemma 34 for $r$ (which holds since we assume $\eta \leq \frac{\gamma}{\mu L}$) we get the divergence bound:

$$D_{\alpha'}(X_{0:Tk}, X'_{0:Tk}) \leq \frac{3\mu\tau\alpha' L^2 c^2(\tau d + \ln(\frac{1}{\delta_1}) + \ln(\frac{1}{\delta_2}))\eta^2}{4}$$

We can then just follow the proof of Theorem 9 as long as:

$$c_1(\alpha') = \exp\left(\frac{3\mu\tau^2 d\alpha'(\alpha'-1)L^2 c^2 \eta^2}{4}\right) < 2,$$

For $\alpha' = \frac{12\alpha \ln \alpha \ln 1/\varepsilon}{(\alpha-1)\varepsilon} - 2$. This follows if $\eta = \tilde{O}(\frac{1}{L\tau\mu\ln\alpha} \cdot \frac{\varepsilon}{\sqrt{d}})$ as assumed in the lemma statement. $\square$

## E.2 Proof of Lemma 34

*Proof.* We can assume $\delta < 1/2$, at a loss of a multiplicative constant. We first focus on the continuous chain. It suffices to show the maximum norm of the velocity over $[0, \tau)$ is $v_{\max}$ with the desired probability. We will instead focus on bounding the Hamiltonian, defined as follows:

$$\phi_t = \mu f(x'_t) + \|v'_t\|_2^2/2.$$

Analyzing the rate of change, by Ito's lemma we get

$$\begin{aligned}
d\phi_t &= \frac{\partial \phi_t}{\partial x'_t} \cdot dx'_t + \frac{\partial \phi_t}{\partial v'_t} \cdot dv'_t + \frac{1}{2}\left[\sum_{i,j\in[d]} \frac{\partial^2 \phi_t}{\partial(v'_t)_i \partial(v'_t)_j} \frac{d(v'_t)_i}{dB_t} \frac{d(v'_t)_j}{dB_t}\right] dt \\
&= \mu\nabla f(x'_t) \cdot v'_t dt + v'_t \cdot (-\mu\nabla f(x'_t)dt - \gamma v'_t dt + \sqrt{2\gamma\mu}dB_t) + 2\gamma\mu d \cdot dt \\
&= \gamma(2\mu d - \|v'_t\|_2^2)dt + \sqrt{2\gamma\mu}(v'_t \cdot dB_t).
\end{aligned}$$

So, we can write the Hamiltonian at any time as a function of the initial Hamiltonian $\phi_0$ and the random variables $B_t$ and $v'_t$ as:

$$\phi_t = \phi_0 - \gamma\int_0^t \|v'_s\|_2^2 ds + \sqrt{2\gamma\mu}\int_0^t \|v'_s\|_2 \frac{v'_s}{\|v'_s\|_2} \cdot dB_s + 2\gamma\mu dt.$$

Let $V_t$ denote $\int_0^t \|v'_s\|_2^2 ds$. By scalability of Brownian motion, we can define a Brownian motion $B'_t$ jointly distributed with $B_t$ such that $dB_t = \frac{1}{\|v'_t\|_2} \frac{d}{dt}\int_0^{V_t} dB'_s$. Then, we have:

$$\phi_t = \phi_0 - \gamma V_t + \sqrt{2\gamma\mu}\int_0^{V_t} \frac{v'_{g(s)}}{\|v'_{g(s)}\|_2} \cdot dB'_s + 2\gamma\mu dt,$$

Where $g(r)$ is the value $r'$ such that $\int_0^{r'} \|v'_s\|_2^2 ds = r$. We can then use the rotational symmetry of Brownian motion to define another Brownian motion $B''_t$ jointly distributed with $B'_t$ such that $u \cdot dB''_t = \frac{v'_{g(t)}}{\|v'_{g(t)}\|_2} \cdot dB'_t$ for a fixed unit vector $u$, giving:

$$\phi_t = \phi_0 - \gamma V_t + \sqrt{2\gamma\mu}\int_0^{V_t} u \cdot dB''_s + 2\gamma\mu dt.$$

We will show that with probability at least $1 - \delta$ over $B_t''$, the maximum of $\phi'(V) := \phi_0 - \gamma V + \sqrt{2\gamma\mu} \int_0^V u \cdot dB_s''$ over $V \in [0, \infty)$ is at most $\frac{1}{4} v_{\max}^2$. Under this event, if $c$ is sufficiently large then for all $t \in [0, \tau)$ we have $\phi_t \leq \frac{1}{4} v_{\max}^2 + 2\gamma\mu d\tau \leq \frac{1}{2} v_{\max}^2$, giving the desired velocity bound.

We first claim that with probability at at least $1 - \frac{\delta}{2}$. for all non-negative integers $k$, we have $\phi'(kv_{\max}^2) \leq -\frac{(k-1)v_{\max}^2}{2}$. For sufficiently large $c$, this holds for $k = 0$ with probability at least $1 - \frac{\delta}{4}$ by (4). Conditioning on this event, for $k > 0$ if $\phi'(kv_{\max}^2) \geq -\frac{(k-1)v_{\max}^2}{2}$, then:

$$\sqrt{2\gamma\mu} \int_0^{kv_{\max}^2} u \cdot dB_s'' = N(0, 2k\gamma\mu v_{\max}^2) \geq -\frac{(k-1)v_{\max}^2}{2} - \phi_0 + k\gamma v_{\max}^2 \geq (\gamma - 1)kv_{\max}^2,$$

Which occurs with probability at most $\exp(-\frac{(\gamma-1)^2 k^2 v_{\max}^4}{4k\gamma\mu v_{\max}^2}) \leq \exp(-\frac{kv_{\max}^2}{8\mu})$. If the constant $c$ in $v_{\max}$ is sufficiently large, then this is less than $\frac{\delta^{k+2}}{2}$. Taking a union bound over all $k$, we get the claim. Next, we claim that in each interval $[kv_{\max}^2, (k+1)v_{\max}^2)$, the maximum increase of $\phi'(V)$ is more than $(\frac{k+1}{2})v_{\max}^2$ with probability at most $\frac{\delta^{k+2}}{2}$. Taking a union bound over all intervals, this claim along with the previous claim this gives the desired bound on $\phi'(V)$ with probability $1 - \delta$. This claim follows by observing that in the interval $[kv_{\max}^2, (k+1)v_{\max}^2)$, $\phi'(V)$ increases more than $\max_{V \in [kv_{\max}^2, (k+1)v_{\max}^2)} \left[ \int_{kv_{\max}^2}^V u \cdot dB_s'' \right]$, which is at most $(\frac{k+1}{2})v_{\max}^2$ with probability at most $\exp(-\frac{(\frac{k+1}{2})^2 v_{\max}^4}{8v_{\max}^2}) \leq \frac{\delta^{k+1}}{2}$.

The discrete chain is analyzed similarly. We have:

$$d\phi_t = \frac{\partial \phi_t}{\partial x_t} \cdot dx_t + \frac{\partial \phi_t}{\partial v_t} \cdot dv_t + \frac{1}{2} \left[ \sum_{i,j \in [d]} \frac{\partial^2 \phi_t}{d(v_t)_i d(v_t)_j} \frac{d(v_t)_i}{dB_t} \frac{d(v_t)_j}{dB_t} \right] dt$$

$$= \mu \nabla f(x_t) \cdot v_t dt + v_t \cdot (-\mu \nabla f(x_{\lfloor \frac{t}{\eta} \rfloor \eta}) dt - \gamma v_t dt + \sqrt{2\gamma\mu} dB_t) + 2\gamma\mu d \cdot dt$$

$$= \mu(\nabla f(x_t) - \nabla f(x_0)) \cdot v_t dt - \gamma\|v_t\|_2^2 dt + \sqrt{2\gamma\mu}(v \cdot dB_t) + 2\gamma\mu d \cdot dt$$

$$\leq \mu L \|x_t - x_{\lfloor \frac{t}{\eta} \rfloor \eta}\|_2 \|v_t\|_2 dt - \gamma\|v_t\|_2^2 dt + \sqrt{2\gamma\mu}(v \cdot dB_t) + 2\gamma\mu d \cdot dt$$

$$= \mu L \| \int_{\lfloor \frac{t}{\eta} \rfloor \eta}^t v_s ds\|_2 \|v_t\|_2 dt - \gamma\|v_t\|_2^2 dt + \sqrt{2\gamma\mu}(v \cdot dB_t) + 2\gamma\mu d \cdot dt$$

$$\leq \mu L \left( \int_{\lfloor \frac{t}{\eta} \rfloor \eta}^t \|v_s\|_2 \|v_t\|_2 ds \right) dt - \gamma\|v_t\|_2^2 dt + \sqrt{2\gamma\mu}(v \cdot dB_t) + 2\gamma\mu d \cdot dt$$

$$\leq \frac{\mu L}{2} \left( \int_{\lfloor \frac{t}{\eta} \rfloor \eta}^t \|v_s\|_2^2 + \|v_t\|_2^2 ds \right) dt - \gamma\|v_t\|_2^2 dt + \sqrt{2\gamma\mu}(v \cdot dB_t) + 2\gamma\mu d \cdot dt.$$

Integrating, we get:

$$\phi_t \leq \phi_0 - (\gamma - \frac{\mu L \eta}{2}) \int_0^t \|v_s\|_2^2 ds + \sqrt{2\gamma\mu} \int_0^t \|v_s\|_2 \frac{v_s}{\|v_s\|_2} \cdot dB_s + 2\gamma\mu dt$$

$$\leq \phi_0 - \frac{\gamma}{2} \int_0^t \|v_s\|_2^2 ds + \sqrt{2\gamma\mu} \int_0^t \|v_s\|_2 \frac{v_s}{\|v_s\|_2} \cdot dB_s + 2\gamma\mu dt.$$

At this point we repeat the analysis from the continuous case (only losing a multiplicative constant due to the $\gamma/2$ multiplier not being $\gamma$). $\qquad \square$

# F  On Distance Measures between Distributions

Existing algorithms for sampling from logconcave distributions are known to output samples from a distribution that is close to the intended distribution. The closeness is typically measured in statistical

distance, Wasserstein distance, or in KL divergence. Unfortunately, none of these distances are strong enough to ensure differential privacy for the resulting algorithm. The more stringent choice of distance in differential privacy is for a good reason: it is easy to construct examples of algorithms that ensure privacy with respect to one of these weaker notions of distance but are clearly unsatisfactory from a privacy point of view [Dwork and Roth, 2014]. This motivates the question of efficient sampling in terms of a stronger measure of distance such as $\infty$-divergence, or Rényi divergence (both of which upper bound KL divergence and thus upper bound statistical distance and Wasserstein distances). Different distance notions can be related to each other and Hardt and Talwar [2010] showed that an exponentially small statistical distance guarantee suffices to derive a differentially private algorithm. This allows for polynomial time algorithms using the classical logconcave samplers.

The faster sampling algorithms based on Langevin dynamics and relatives however have a polynomial dependence on the distance. In this case, convergence under the various notions of distance is not equivalent. None of the commonly used measures (Statistical distance, KL-divergence or Wasserstein distance) can be polynomially related to common distances of interest from a privacy point-of-view ($\infty$-divergence, Rényi divergence). While $(\varepsilon, \delta)$-DP can be related via a polynomial in $\delta^{-1}$, this would lead to algorithms that are polynomial in $\delta^{-1}$, which is undesirable as we often want $\delta$ to be sub-polynomial.

## Footnotes

[4]In particular, recalling the proof of Facts 26 and 27, we can write $A$ explicitly as $\lim_{k \to \infty} \prod_{j=0}^{k-1} (I_d - \frac{\eta}{k} \nabla^2 f(z_j))$, where $z_j$ is some point on the path from 0 to $x'_{t+\frac{j\eta}{k}}$. Each $A_s$ can be written similarly, except only considering the gradient descent process from time $t + s$ to $t + \eta$.