[Reviews · NeurIPS 2020]

Review 1

Summary and Contributions: In this paper, the authors discuss the role of approximate sampling and its impact on privacy of the algorithm. In particular, the authors study the problem of sampling from distributions with density proportional to exp(-f). Under conditions on f, authors establish that discretized overdamped Langevin dynamics converges to the target density in Renyi divergence which is the suitable notion of "distance" for Differential Privacy. Authors achieve this by showing that the discretized Langevin dynamics is close in Renyi Divergence to the continuous-time dynamics and then apply the result from Vempala and Wibisono [2019] that had established the convergence of continuous-time Langevin dynamics.

Strengths: The problem studied in the paper is an important one and is very relevant to the differential privacy community at Neurips. As suggested in the paper, running exponential mechanism on continuous state space requires approximate sampling. One needs to design approximate samplers that still preserve privacy. In this paper authors provide theoretical guarantees for discretized langevin algorithm.

Weaknesses: One weakness I foulnd with this work is that the paper focusses on algorithms preserving differential privacy but authors authors don't exactly establish how the privacy is preserved. This is almost entirely left as an excercise for the reader. I would suggest that the authors should add a result that explains how the privacy parameter of the algorithm is preserved. Another question would be if the original sampler preserved (\veps-\delta) differential privacy, does this privacy parameter change after approximating with the proposed discretized langevin algorithm.

Correctness: The claims in the paper seem logical. From only briefly skimming the proofs in the appendix, I couldn't find serious flaws.

Clarity: The paper is well written on its own.

Relation to Prior Work: This work primarily builds on top of the convergence results of continuous-time Langevin dynamics established by Vempala and Wibisono [2019]. But there has been prior work that studies MCMC algorithms that maintains differential privacy. In particular, authors should consider also citing: Heikkilä, Mikko, et al. "Differentially private markov chain monte carlo." Advances in Neural Information Processing Systems. 2019.

Reproducibility: Yes

Additional Feedback: ======= Post Rebuttal Comments ======= I thank the author for clarifying the issues I had raised about how the privacy is preserved. After reading through the authors feedback and other reviews I am convinced that this is a good submission.


Review 2

Summary and Contributions: This paper analyzes the convergence of discrete-time Langevin dynamics to its stationary distribution (the Gibbs distribution \propto e^{-f(x)}) when f is strongly convex and smooth function f; and the convergence to continuous-time Langevin dynamics when f is Lipschitz and Smooth (potentially non-convex). Different from previous approaches that establish convergence in TV-distance, the current paper addresses the same problem under the stronger Renyi-divergence hence substantially improves the computational guarantee of samplers that run Langevin dynamics. This problem is of critical interest to the problem of differentially private learning via posterior sampling. If the additional technical hurdle that I discuss in “Weaknesses” can be resolved, this paper has then provided a satisfactory solution to the long-standing open problem of analyzing the differential privacy of running Langevin dynamics for a finite number of iterations via convergence to the stationary distribution, rather than via composition. The number of iterations needed is linear in dimension d and has no polynomial dependence in 1/\delta, which is a big improvement from existing work. The paper also makes a number of interesting technical contributions including using tools from Renyi-DP in novel ways for analyzing the convergence for bounding the gap between the discrete-time Langevin dynamics and its continuous time counterpart. Overall, I think this is an excellent paper that will get on the reading list of many researchers in the differential privacy community.

Strengths: 1. Well-motivated research problem. 2. Strong results with both theoretical and practical interest 3. The presentation is smooth

Weaknesses: 1. Despite hinting at such a result multiple times in the paper, the results presented in this paper does not directly imply pure or approximate differential privacy for an algorithm that runs Langevin dynamics for T-iterations. At least it is not a trivial argument that goes through without further assumptions. The reason is the following: The Renyi Divergence bound on D(P||R) (the order \alpha is abbreviated for readability) does not seem to imply a differential privacy bound overall, even though a sample from R satisfies DP. The DP bound of posterior sampling implies a bound on D(R||R’). By results of this paper, we have bounds on D(P||R) and D(P’||R’). However even as D(P||R) and D(P’||R’) both converging to 0, there might not be any bound on D(P||P’). As an example, R = R’ is uniform [0,1], P_t is uniform [0,1-1/t]. P’_t is uniform [0,1-1/sqrt(t)], and let t = 2,3,4,5,… In this case, D(R||R’) = 0, D(P_t||P’_t) = +\infty for all t even though D(P||R) and D(P’||R’) converge to 0. It is unclear whether such transitive properties can be proven in the special case of interest to this paper. You need a bound on the renyi-divergence of D(R || P) too I think, which is not impossible intuitively because f is 1-strongly convex thus e^{-f} dominates P in the part of P that is purely Gaussian (outside the sum of gradients after T steps). So having P in the denominator is potentially “easier” than bounding D(P||R). 2. The paper aims at showing that you can run discrete-time Langevin dynamics forever and as it converges to the stationary distribution just like the continuous-time Langevin dynamics, it avoids the composition-argument that is used before, which are in some sense, a crude and wasteful way of analyzing the privacy-property of this method when only the final iterate is released. Interestingly, however, the argument that the authors used to establish such a bound applies only for the case when running a discrete-time Langevin dynamics for a fixed number of times. You cannot really fix a step size and then let the discrete-time Langevin dynamics indefinitely. This is not really a weakness because finite time bound is more useful in practice, but it is like it bypassed, rather than directly solved, the problem stated in the beginning.

Correctness: I checked the proofs as much as I can. Specifically, I checked the proof of Lemma 5, Lemma 7 Lemma 9 carefully. And then I checked the high-level steps of Theorem 8, Theorem 1 and Theorem 12. I think the proofs are correct (even though the results do not solve the motivating problem of getting differential privacy for finite steps of Langevin dynamics).

Clarity: The paper is clearly written.

Relation to Prior Work: Discussion of the prior work is appropriate.

Reproducibility: Yes

Additional Feedback: 1. The results hinge upon the convergence of continuous time Langevin dynamics to the gibbs distribution in Renyi divergence, which the paper cites existing results saying that it is true under “mild conditions”. It will make the presented results more rigorous if these “mild conditions” are checked carefully in some concrete contexts, e.g., when f is the negative log-likelihood of a generalized linear model. In this way, additional conditions from there could be made explicit. 2. In practice, people run stochastic gradient-based Langevin dynamics instead. This method is known to be differentially private by composing subsampled-gaussian mechanisms which RDP analysis is known. I am curious whether the part of the analysis that connects discrete time SDE to continuous time SDE based on composing the Renyi divergence bounds can be extended to the case when the discrete time SDE is coming from SGLD. That would make it quite interesting. Of course, this is probably beyond the scope of the current paper. --------------------- post-rebuttal ------------------------ I am glad that the authors acknowledged the issue of the Renyi-divergence D(R||P) or D(R||Q) and provided a detailed sketch on how it can be bounded by leveraging the exponential convergence. The discussion is sufficiently detailed and comes with a comment on how it isn't possible for large \alpha. This is important because you need \alpha to be sufficiently large to get meaningful \epsilon parameters in DP. That said, I am happy with the paper even if this problem is not solved.


Review 3

Summary and Contributions: The exponential mechanism is a building block for differentially private algorithms and requires sampling from the distribution exp(-f) for a loss function f. This paper studies fast (computationally efficient) sampling, for the setting where the domain of the distribution is high-dimensional. The authors give three algorithms and convergence rates for sampling from exp(-f) with running time which grows linearly in (resp., with the square-root of) the dimension d and in 1/zeta^2 (zeta=privacy parameter), for smooth and strongly convex (resp., Lipschitz) functions f.

Strengths: -The use of the exponential mechanism is ubiquitous in differentially private algorithms and more often than not, are computationally inefficient. Prior work has given sampling algorithms with polynomial running time with respect to the dimension but this is still inefficient for high dimensional settings. So these fast sampling algorithms (which have d or sqrt{d} running time) would make these methods more practical. -These algorithms significantly improve over prior running times (to the best of my knowledge, even for bounded domains). -The results are based on the analysis of the convergence rate of Langevin dynamics with respect to the Renyi divergence, which is novel (as previous convergence analyses for these algorithms were with respect to weaker distances, which are not strong enough to imply closeness in the more strict sense of differential privacy).

Weaknesses: I consider this paper a complete piece of work. Answers to related questions such as similar bounds for more general functions f, or whether these bounds can be improved, are interesting directions but I do not see the lack of them as a weakness of this work.

Correctness: The claims of the paper are fully proven.

Clarity: I found the paper very well-written and that it had a very good split/connection between the intuition in the main body and the technical content in the supplementary.

Relation to Prior Work: Relation to prior work is sufficiently discussed. Although the discussion is thorough, I think an exact bound of the best known results for some interesting cases (e.g. bounded domain?) should be added for comparison, so that the results of Figure 1 can be more directly appreciated.

Reproducibility: Yes

Additional Feedback: As I mentioned, I found the presentation of the paper very good. My only suggestion would be the comment above to allow for more direct quantitative comparison with related work. I almost did not find any typo in this manuscript! -Lemma 6: one of the two X_0 should be a X_0' ===================================== Thank you for your response. I originally missed the fact that the second part of the privacy guarantee was not proven. However, the proof sketch is sufficiently detailed to support the result stated in the response. I will keep my score as is and still support acceptance of this paper, anticipating that you can add this proof as well as a discussion on the privacy parameter epsilon that is achievable.


Review 4

Summary and Contributions: For a smooth and strongly convex f, the target is to sample from the distribution $\exp(-f)$. This paper gives convergence analysis on the difference between the distribution from a discretized Langevin process and the distribution from a continuous process under Renyi divergence.

Strengths: This paper is mainly on theoretical analysis. The conclusions are solid. The paper borrows tools used in differential privacy.

Weaknesses: 1. The importance of Renyi divergence is not well explained. It is not clear why Renyi divergence is so special for a reader who is not familiar with differential privacy. 2. What is the benefit using the unadjusted Langevin process, instead of the adjusted process? If we use the adjusted process, we may get the exact distribution. 3. Is the obtained rate optimal? How far is it from the optimal rate? At least it is not a good rate comparing with the results from KL divergence. 4. The paper does not provide any new algorithm or new insight about the algorithm. It is just a theoretical analysis on existing algorithm under a difference divergence. 5. Can you hightlight the novelty in proofs? What new techniques are developed and necessary in proofs? 6. Is it possible to show some numerical results to demonstrate that the theoretical results are aligned with the numerical results?

Correctness: The claims are correct. There is no empirical study.

Clarity: 1. There are some notation in Section 2 without any introduction, such as D_\alpha, r. There are introduced in later sections. 2. What is \perp in line 187?

Relation to Prior Work: By changing from KL, Wasserstein divergence to Renyi divergence, what mathematical techniques are introduced to establish the results?

Reproducibility: Yes

Additional Feedback:

[Author Response · NeurIPS 2020]

We are grateful to the reviewers for their insightful reviews and feedback. We have incorporated fixes to simple issues
such as typos and missing references and do not address those issues here.

Reviewer 1, 2, and 4 commented that is that it is not clear how the Renyi divergence bounds in the paper relate/translate
to private samplers. An additional issue that Reviewer 2 points out is that while we show that $D_\alpha(P||R)$ is small for $P$
being the distribution of the discrete finite-time dynamics and $R$ being the stationary distribution, we need to also show
that $D_\alpha(R||P)$ to get efficient private samplers. We address this below. Assuming that we can bound both $D_\alpha(P||R)$
and $D_\alpha(R||P)$, the conversion to $(\epsilon, \delta)$-DP follows from the following argument. Suppose the underlying mechanism
guarantees that $R, R'$ satisfy $(\epsilon, \delta)$-differential privacy. Fact 9 shows that if for $\alpha = 1 + 2\ln(1/\delta)/\epsilon$, $D_\alpha(P||R)$ and
$D_\alpha(R||P)$ are both at most $\epsilon/2$, then $P, R$ also satisfy the (bi-directional) divergence guarantee of $(\epsilon, \delta)$-differential
privacy (and the same for $P', R'$). Then by composition theorems, $P, P'$ satisfy $(3\epsilon, 3\delta)$-differential privacy. In our
revisions to the paper, we will include a formal theorem/proof for this argument.

Reviewer 2 points out that our result "bypasses" the problem of bounding the bias of the discrete dynamics' stationary
distribution. This is indeed the case and we discuss this briefly in our introduction. This can be seen as both a strength
and weakness of our approach. We will also add this as an interesting future direction.

Reviewer 2 suggests making the results more rigorous by verifying the "mild conditions" under which Langevin
dynamics converges. The result of Vempala and Wibisono suggests that a sufficient condition is strong convexity, which
we assume in the paper. This will be made more explicit in the introduction.

Reviewers 3 and 4 suggest further comparisons of our iteration complexity to previous results and clarifying whether
the result is optimal or near-optimal. We do currently remark in our concluding section that our dependence on $\epsilon$ is
worse than that of results for KL-divergence, and also point out that a factor of $1/\epsilon^2$ seems inherent to our analysis.
Our revision will this discussion our introduction to make these points clearer to the reader. Unfortunately, we do not
know of any lower bounds for the iteration complexity needed for Langevin dynamics to converge in Rényi divergence,
so we are unable to comment on the optimality of the main result. We do however note that for Differential Privacy
applications, the desired $\epsilon$ is usually a small constant (say 0.1 or 0.5).

Reviewer 4 asks the benefit of using the unadjusted Langevin process. Part of our goal is to provide a simple and more
accessible analysis, and our analysis is most simple when adapted to this algorithm. However, our approach might be
useful to bound the discretization error of algorithms with Metropolis steps.

Reviewer 4 asks to highlight the novelty of the proofs. As mentioned in the introduction, we feel the novelty is the
simplicity of the approach, requiring almost no stochastic calculus, making the analysis ideally more accessible to e.g.
members of the differential privacy community who are less familiar with the stochastic calculus literature.

We end the rebuttal with a discussion on bounding $D_\alpha(R||P)$, that we overlooked in the submission. Technically, the
proof is essentially identical to $D_\alpha(P||R)$ case, and we give details for completeness. The results in the submission
bound both $D_\alpha(P||Q)$ and $D_\alpha(Q||P)$, where $Q$ is the distribution of the continuous finite-time dynamics. While we
state Theorem 8 as bounding $D_\alpha(P||Q)$, we provide a bi-directional divergence bound in Lemma 5, and so the proof
of Theorem 8 easily also bounds $D_\alpha(Q||P)$. What remains is to bound $D_\alpha(R||Q)$, since the cited paper of Vempala
and Wibisono (VW19) only immediately bounds $D_\alpha(Q||R)$. However, the techniques in that paper can easily be
generalized to also derive a bound $D_\alpha(R||Q)$. Due to space constraints we can only provide a high-level summary of
this generalization here, but for completeness our revisions include a full explanation in the appendix.

We need two ingredients from VW19 to bound $D_\alpha(R||Q)$. The first is to show that if our initial distribution $Q_0$ is a
Gaussian with the correct variance then ideally $D_\alpha(Q_0||R), D_\alpha(R||Q_0) \lesssim d$ (as opposed to Lemma 9 in our paper and
the cited lemma from VW19, which only shows $D_\alpha(Q_0||R) \lesssim d$). This guarantee is not attainable for large $\alpha$. However,
for $Q_0 = N(0, I)$ and any 1-smooth and $L$-strongly convex $f$, both divergences are bounded by $d$ if, say, $\alpha = 1 + 1/2L$.
The Hypercontractivity Lemma (Lemma 14) in VW19 shows that after running for continuous time $t$ proportional to
$\log(\alpha' L)$ (much smaller than the current continuous time bound we use), we will get that $D_{\alpha'}(R||Q_t), D_{\alpha'}(Q_t||R)$ are
both also finite and roughly $d$. The choice of $Q_0 = N(0, I)$ also satisfies the properties stated in Lemma 5 of our paper.

The second is to show that $D_\alpha(R||Q)$ decreases exponentially. We show that the continuous chain (as opposed to the
stationary distribution) always satisfies LSI with constant 1/3, and then we can slightly modify Lemma 5 and 6 in
VW19 to show that $D_\alpha(R||Q)$ decays exponentially. Our choice of initial distribution $Q_0$ satisfies LSI with constant 1.
The continuous Langevin dynamics can be viewed as the limit of the discrete chain, which is repeated application of
a $(1 - \eta/2)$-Lipschitz gradient step followed by adding Gaussian noise $N(0, 2\eta I)$, as $\eta$ approaches 0. Lemma 16 in
VW19 shows that applying a $(1 - \eta/2)$-contractive map increases the LSI constant by at least a multiplicative factor
of $(1 - \eta/2)^{-2}$, and adding $N(0, 2\eta I)$ changes the LSI constant from $\alpha$ to at least $\frac{1}{1/\alpha+2\eta}$ (see e.g. and "Functional
Inequalities for Convolution Probability Measures" by Wang and Wang). Taking the limit as $\eta$ goes to zero, we get that
the continuous Langevin dynamics do not cause the LSI constant to decrease below 1/3.

[Meta-Review · NeurIPS 2020]

The reviewers agree that this paper provides an interesting analysis on the Langevin dynamics, which has interesting implications to differential privacy. The presentation is clear and the technical results are novel. The paper should clarify whether the finite-time variant of the dynamics actually leads to a private algorithm in their revision.